# Single-cell transcriptomics and cell-specific proteomics reveals molecular signatures of sleep

Pawan K. Jha[1,2,5], Utham K. Valekunja[1,2,5], Sandipan Ray[1,2,3], Mathieu Nollet[1,2,4] & Akhilesh B. Reddy [1,2✉]

Every day, we sleep for a third of the day. Sleep is important for cognition, brain waste clearance, metabolism, and immune responses. The molecular mechanisms governing sleep are largely unknown. Here, we used a combination of single-cell RNA sequencing and cell-type-specific proteomics to interrogate the molecular underpinnings of sleep. Different cell types in three important brain regions for sleep (brainstem, cortex, and hypothalamus) exhibited diverse transcriptional responses to sleep need. Sleep restriction modulates astrocyte-neuron crosstalk and sleep need enhances expression of specific sets of transcription factors in different brain regions. In cortex, we also interrogated the proteome of two major cell types: astrocytes and neurons. Sleep deprivation differentially alters the expression of proteins in astrocytes and neurons. Similarly, phosphoproteomics revealed large shifts in cell-type-specific protein phosphorylation. Our results indicate that sleep need regulates transcriptional, translational, and post-translational responses in a cell-specific manner.

[1] Department of Systems Pharmacology & Translational Therapeutics, Perelman School of Medicine, University of Pennsylvania, Philadelphia, PA 19104, USA.
[2] Institute for Translational Medicine and Therapeutics, Perelman School of Medicine, University of Pennsylvania, Philadelphia, PA 19104, USA. [3] Present address: Department of Biotechnology, Indian Institute of Technology Hyderabad, Kandi, Sangareddy 502285 Telangana, India. [4] Present address: Imperial College London, South Kensington Campus, London SW7 2AZ, UK. [5] These authors contributed equally: Pawan K. Jha, Utham K. Valekunja. ✉email: areddy@cantab.net

Sleep is an essential component of daily life. The explicit function of sleep is still a mystery. Insufficient or disturbed sleep is associated with the accumulation of brain waste, cognitive impairments, increased risk of metabolic abnormalities, and suppressed immune responses[1–4]. Restoration of many of these abnormalities by sleep suggests that prolonged wakefulness imposes a load on the system that is alleviated by sleep[2,5–7]. The progressive rise of sleep need during wakefulness, and its dissipation during sleep, maintain sleep homeostasis[8,9].

Sleep homeostasis is an intrinsic process that regulates sleep need by the cumulative and interactive action of multiple brain regions and cells[10–12]. Sleep-wake cycles are regulated mainly in the hypothalamus, brainstem, and cortex of the brain[13,14]. Although prior work has described bulk transcriptional responses of various brain regions following acute sleep restriction[15–19], cell-type-specific changes in gene and protein expression have not been delineated. Thus, there is a gap in our knowledge concerning the cell types that are responsible for transcriptional changes during sleep deprivation, and whether different cell types are affected in different ways by sleep deprivation. Here, we set out to bridge this gap by determining the molecular changes within individual brain cells across sleep-wake states.

We used single-cell RNA sequencing (scRNA-seq) to compare the cellular composition and transcriptomic profiles of the brainstem, cortex, and hypothalamus of mice going through different sleep treatments. For all the major cell populations that are responsive to sleep treatments, we provide a comprehensive census of gene expression and analysis of molecular pathways affected. We show that sleep need regulates the transcriptional profiles of brain cells in the three brain regions differently. Given that astrocyte-neuron signaling plays a crucial role in homeostatic brain function[20], we delineated ligand-receptor interactions that are altered by sleep treatments within these cell types. Our scRNA-seq data enabled us to identify uniquely enriched transcription factors controlling changes in gene expression in the brain cells from sleep-deprived animals. We also analyzed the change in protein levels, and post-translational phosphorylation, following sleep treatments in cortical astrocytes and neurons. Taken together, this study substantially advances our understanding of the molecular changes in sleep homeostasis at the individual brain cell level and serves as a foundation for further cell-specific interrogation of sleep need.

## Results

**Sleep phenotyping and experimental design**. We sleep-restricted wild-type C57BL/6 J mice by prolonging their wakefulness into the light phase of the light-dark cycle, from ZT0 to ZT12 (ZT; *zeitgeber* time). This is when mice normally obtain 60–70% of their daily sleep[21]. We recorded the electroencephalography (EEG) throughout the experiment (Day 1: baseline, Day 2: sleep deprivation and recovery, Day 3: recovery). To characterize sleep loss due to sleep deprivation we analyzed the time spent in sleep and wake, delta, and theta power throughout the time course of the experiment (Fig. 1c, d and Supplementary Fig. 1). Animals had 91% sleep loss in the first six hours of sleep restriction (ZT0-6) and 86% in the later six hours (ZT6-12) (Fig. 1b). As expected, the mice showed increased sleep need, as evidenced by elevated slow-wave activity (SWA) and delta power (1–4 Hz) in their EEG traces in the recovery period after sleep deprivation[8,9] (Fig. 1d and Supplementary Fig. 1d). Despite having a strong circadian activity component during the recovery period (circadian nighttime), we reported a significant rise in NREM sleep from ZT12–18 and NREM delta power from ZT12–14 compared to their corresponding baseline hours (Fig. 1c, d and Supplementary Data 1). Our sleep restriction protocol did not affect the plasma

corticosterone level of the experimental animals (Supplementary Fig. 1e). This indicates accumulated sleep pressure during prolonged wakefulness is independent of stress.

We next divided the mice into three experimental groups: a control normal sleep (NS) group, a 12 h sleep deprived (SD) group, and a sleep deprivation followed by 24 h of recovery sleep (RS) group (Fig. 1a). From each group, we collected samples from three brain regions (brainstem, cortex, and hypothalamus) that are pivotal for regulating sleep-wake cycles[14,22]. Importantly, we controlled for variation due to time of day by harvesting brain tissue at the same circadian phase (ZT12) (Fig. 1a). We then performed single-cell RNA sequencing (scRNA-seq) on cell suspensions from each brain region, in each sleep treatment group (Fig. 2a).

**Identification and transcriptional profiling of brain cells across sleep-wake states**. To understand how the change in sleep-wake states affects gene expression patterns within the cells of all three brain areas, we transcriptionally profiled cells and visualized them using a nonlinear dimensionality-reduction method, uniform manifold approximation and projection (UMAP) (Fig. 2b, c)[23]. Considering all three experimental sleep groups and the three brain regions together, we generated clusters from the quality criteria passed 29,051 cells (Supplementary Fig. 2a–c). When we analyzed each brain region individually, we found that individual cells were distributed among distinct clusters for each brain region, and there was no variation in these clusters for different sleep treatments (Fig. 2c). This indicates there was no global shift in the molecular identity of cells in each brain region during sleep deprivation and recovery, which was as expected for terminally differentiated cells in the brain.

We next used UMAP to subdivide the globally defined clusters within the brain regions and used these subclusters to identify cell types based on known markers derived from existing single-cell data, and the available literature[24–27] (Figs. 3a, b, 4a, b, 5a, b and Supplementary Fig. 2d–f, and Supplementary Table 1). We classified 10,521 cells into 18 clusters defining 12 cell types within the brainstem, 8451 cells into 14 clusters defining 10 cell types within the cortex, and 10,081 cells into 19 clusters defining 13 cell types within the hypothalamus. Cell types included astrocytes, endothelial cells, ependymocytes (EPC), hemoglobin-expressing vascular cells (HbVC), macrophages, microglia, neurons (Glutamatergic, GABAergic, Inhibitory, and Other), oligodendrocytes (Oligo), pericytes and vascular smooth muscle cells, arterial (VSMCA) (Supplementary Fig. 2d–i).

Next, to understand how the change in sleep need affects transcriptional profiles of cells in the brainstem, cortex, and hypothalamus, we performed differential gene expression analysis of the three sleep treatment groups for each cell type we identified (Figs. 3c–e, 4c–e, 5c–e). We found distinct patterns of transcriptional response to sleep need in the astrocytes, neurons, endothelial cells, and microglia of all three brain regions. Pericytes, VSMCAs and EPCs of the brainstem and hypothalamus also showed similar changes. Interestingly, compared to neurons, astrocytes showed larger alterations in the brainstem and hypothalamus, despite higher proportions of cells identified as neurons in the hypothalamus (Supplementary Table 2 and 3). Of note, there was not a large change in gene expression patterns of Oligo, HbVC, and macrophage cell types following sleep treatments (Figs. 3c–e, 4c–e, 5c–e).

We then determined the overlap of sleep-related alterations between previous bulk cell RNA-seq studies and our scRNA-seq data to assign cell types for those genes[28–30]. We were thus able to assign some expression changes found previously in bulk RNA profiling to astrocytes, neurons, endothelial cells, and microglia in

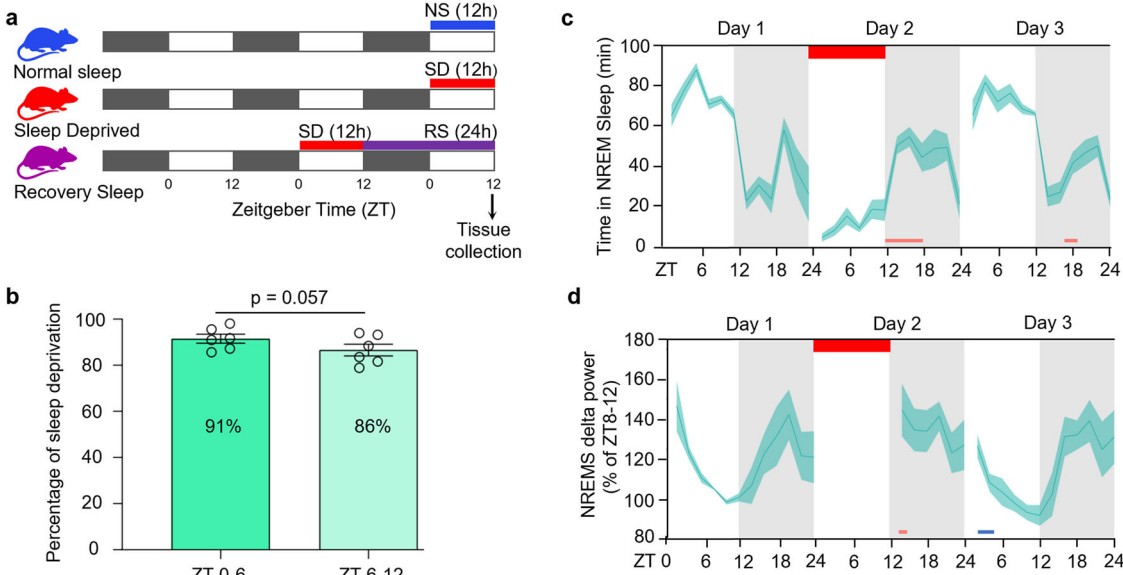

**Fig. 1 Study design and EEG validation of sleep phenotype during sleep deprivation experiment. a** Experimental timeline and tissue collection schedule (black bars represent darkness, white bars indicate periods of light). **b** Percentage of wakefulness during the first half (ZT0-6) and the second half (ZT6-12) of sleep deprivation. **c, d** Ribbon plots showing the time course of 2 hourly binned values of (**c**) NREM state and (**d**) NREM delta (δ)-power percentage of ZT8-12 from Day 1. Horizontal red bar represents the timeline of sleep deprivation (Day 2, ZT 0-12; 12 h duration). Pink and blue bars under the plots denote significant increase and decrease, respectively, compared to Day 1 ($P < 0.05$). Data are mean ± s.e.m. $n = 6$ biological replicates. Paired $t$-test, two-tailed (**b**). Repeated-measures two-way ANOVA followed by post-hoc Fisher's Least Significant Difference (LSD) test (**c** and **d**). For sample sizes and statistics, see Supplementary Data 1.

the cortex and hypothalamus using our novel single-cell data (Supplementary Fig. 3a, b and Supplementary Table 4).

We next performed gene-annotation enrichment analysis of genes differentially expressed across sleep treatments in the cells of the brainstem, cortex, and hypothalamus (Supplementary Figs. 4, 5). In astrocytes, we found significant alterations in genes involved in ribosomal assembly, ribosomal RNA (rRNA) processing, and transmembrane transporter activity. In contrast, vesicle-mediated transport in synapses and nucleotide/nucleoside phosphate metabolism were altered in neurons (Supplementary Figs. 4a, b & 5a, b).

Cell types in different brain areas did not respond to sleep deprivation in the same way. For example, cortical neurons favored genes associated with kinesin binding (and thus anterograde synaptic vesicle transport towards dendrites), whereas hypothalamic neurons responded by altering syntaxin-1 and calcium-dependent protein binding, consistent with their role in vesicle exocytosis (Supplementary Fig. 4b). In contrast, brainstem neurons displayed alterations in genes associated with cytochrome c oxidase activity, suggesting that mitochondrial energy balance is a key process altered in this area. These data suggest distinct functions of sleep in each brain region.

Interestingly, even other "support" cells in the brain exhibited region-specific functional changes. Endothelial cells showed enrichment of protein folding, and epithelial cell proliferation and migration terms, whereas microglia showed changes in genes linked to neuroinflammatory responses and glial cell activation (Supplementary Fig. 5c, d). We also found that EPCs, pericytes, and VSMCAs exhibited significant alterations in annotations associated with protein folding, cellular transition, metal ion homeostasis, and amine metabolism (Supplementary Fig. 5e–g). Some of these annotations (e.g., protein folding and neuroinflammatory response) have been reported in previous sleep studies mapping bulk gene expression changes in the brain[17,31,32]. Overall, these results show that alteration of sleep need causes profound and functionally distinct gene expression changes

within individual cells residing in different brain regions. Our results provide a resource to interrogate sleep-associated gene expression changes in the cells of the brainstem, cortex, and hypothalamus[33,34].

**Validation of transcriptional changes occurring during sleep treatments.** Next, we independently validated our sequencing results by using RNAscope in situ hybridization assays (Fig. 6a). We analyzed mRNA counts per cell in the cortex and hypothalamus of mice brains subjected to sleep treatments. Both sequencing and in situ hybridization results reveal that sleep deprivation enhances the expression of metallothionein-1 (*Mt1*), TSC22 domain family, member 3 (*Tsc22d3*), and gap junction protein, beta 6 (*Gjb6*) in the cortex, and expression levels decrease after recovery sleep (Fig. 6b–d). Similarly, in the hypothalamus, sleep deprivation increases the expression of *Mt1*, but moderately decreases ATPase, Na$^+$/K$^+$ transporting, beta 2 polypeptide (*Atp1b2*), and these levels are further decreased following recovery sleep (Fig. 6e, f). The similar gene expression patterns across sleep treatments revealed by single-cell sequencing and RNAscope thus validate the sequencing results (Fig. 6).

**Sleep need alters astrocyte-neuron interactions.** Astrocyte-neuron signaling is a key player in the maintenance of homeostatic brain function because of coupling between these cell types[20]. This is likely to be highly relevant in the context of sleep homeostasis[35,36]. We, therefore, used our single-cell transcriptomics data to investigate sleep-related changes in astrocyte-neuron communication. We built comprehensive interactive networks of ligands from astrocytes, and receptors expressed in neurons, and vice versa, across all sleep treatment groups in the brainstem, cortex, and hypothalamus (Fig. 7a–f, and Supplementary Fig. 6a–g). We found significant network interactions for Apoe (Fig. 7a–f) and Pomc signaling (Fig. 7d, e), which are key communication nodes in the hypothalamus. *Apoe* was also

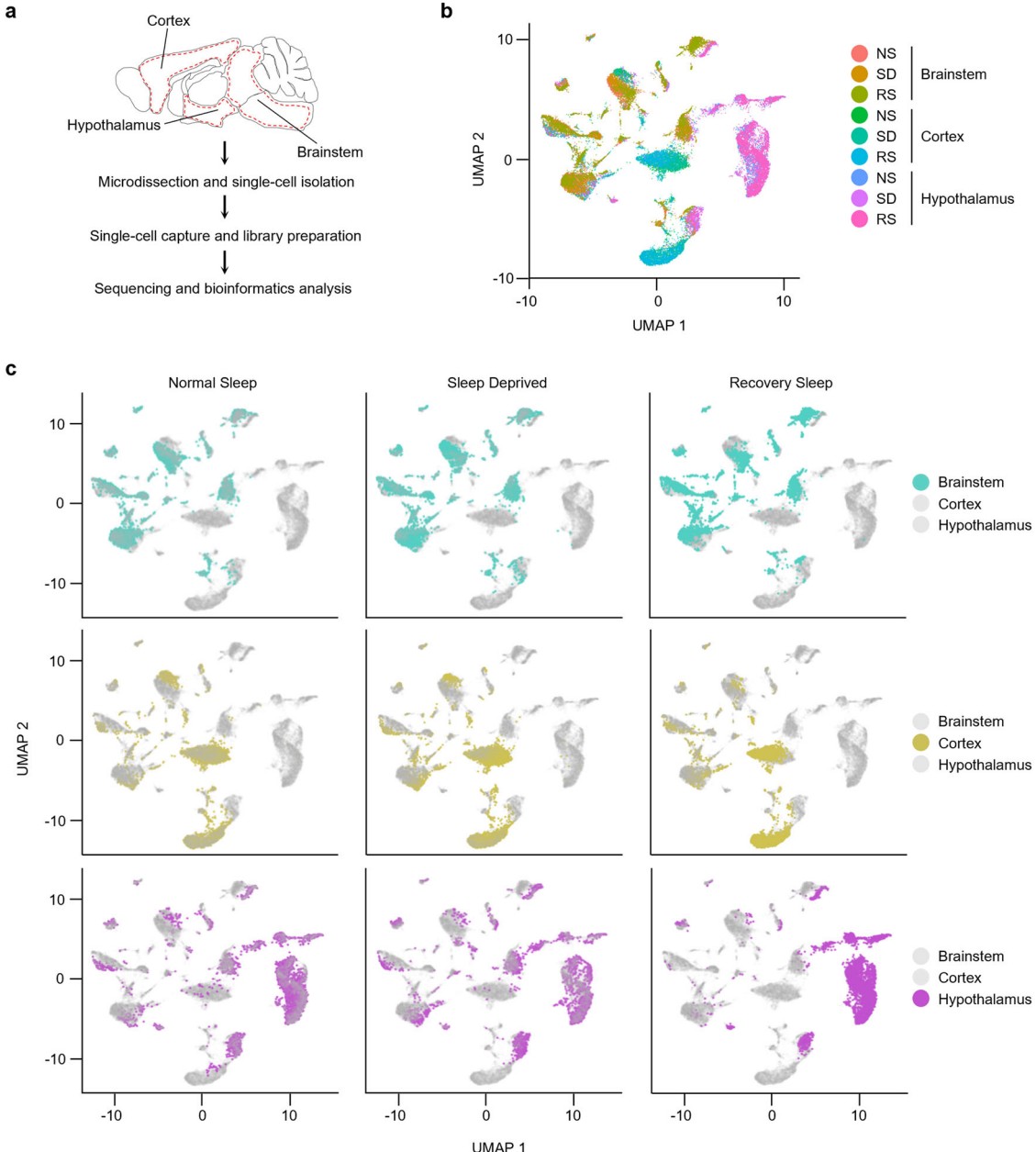

**Fig. 2 Transcriptional profile of brainstem, cortex, and hypothalamic cells across sleep-wake states. a** Workflow for single-cell RNA-seq of cells from brainstem, cortex, and hypothalamus, from microdissection to bioinformatics analysis. Dotted red line indicates areas of each brain region dissected for single-cell sequencing. **b, c** UMAP visualization of 29,051 cells. Cells from Normal Sleep (NS), 12 h Sleep Deprivation (SD), and 12 h Sleep Deprivation followed by 24 h Recovery Sleep (RS) treatment groups from all three brain regions are clustered together (**b**) and separately highlighted (**c**).

upregulated in the brainstem following sleep deprivation (Supplementary Fig. 6a). Interestingly, these genes are linked to sleep consolidation, cognition, and Alzheimer's disease[37,38]. We also found that BDNF signaling is a mediator of crosstalk between neurons and astrocytes in the cortex (Supplementary Fig. 6f, g). Again, the role of *Bdnf* in sleep, insomnia, and stress-related disorders has been documented[39]. Collectively, these results suggest that specific routes of intercellular crosstalk between astrocytes and neurons are important in sleep regulation.

**Sleep modulates gene regulatory processes**. To examine how sleep state controls gene expression changes, we used Single-cell regulatory network inference and clustering (SCENIC) to determine regulons that might coordinate expression patterns[38]. We first evaluated regulon activity scores to profile cells under normal

sleep and sleep-deprived conditions. We then used regulon target information to sort transcription factors (TFs) with their sets of co-expressed target genes (Fig. 7g–i). When we performed gene-annotation enrichment analysis of co-expressed target genes (Supplementary Table 5), we determined different functional annotations predominated in each brain region. The brainstem showed negative regulation of macromolecule biosynthesis (mediated by Sox2) and RNA metabolic processes (Foxj1) (Fig. 7g). In contrast, the cortex expresses TFs that regulate biological rhythms (Hlf) and haemopoiesis (Cebpb) (Fig. 7h). Interestingly, these functions have been reported in recent sleep studies[40,41]. Unlike other brain areas, we identified hypothalamic TFs that regulate calcium ion homeostasis (Atf3), cell death (Fosb), and muscle system process (Mef2c) (Fig. 7i). Recent evidence suggests that Mef2c plays a role in sleep regulation[42].

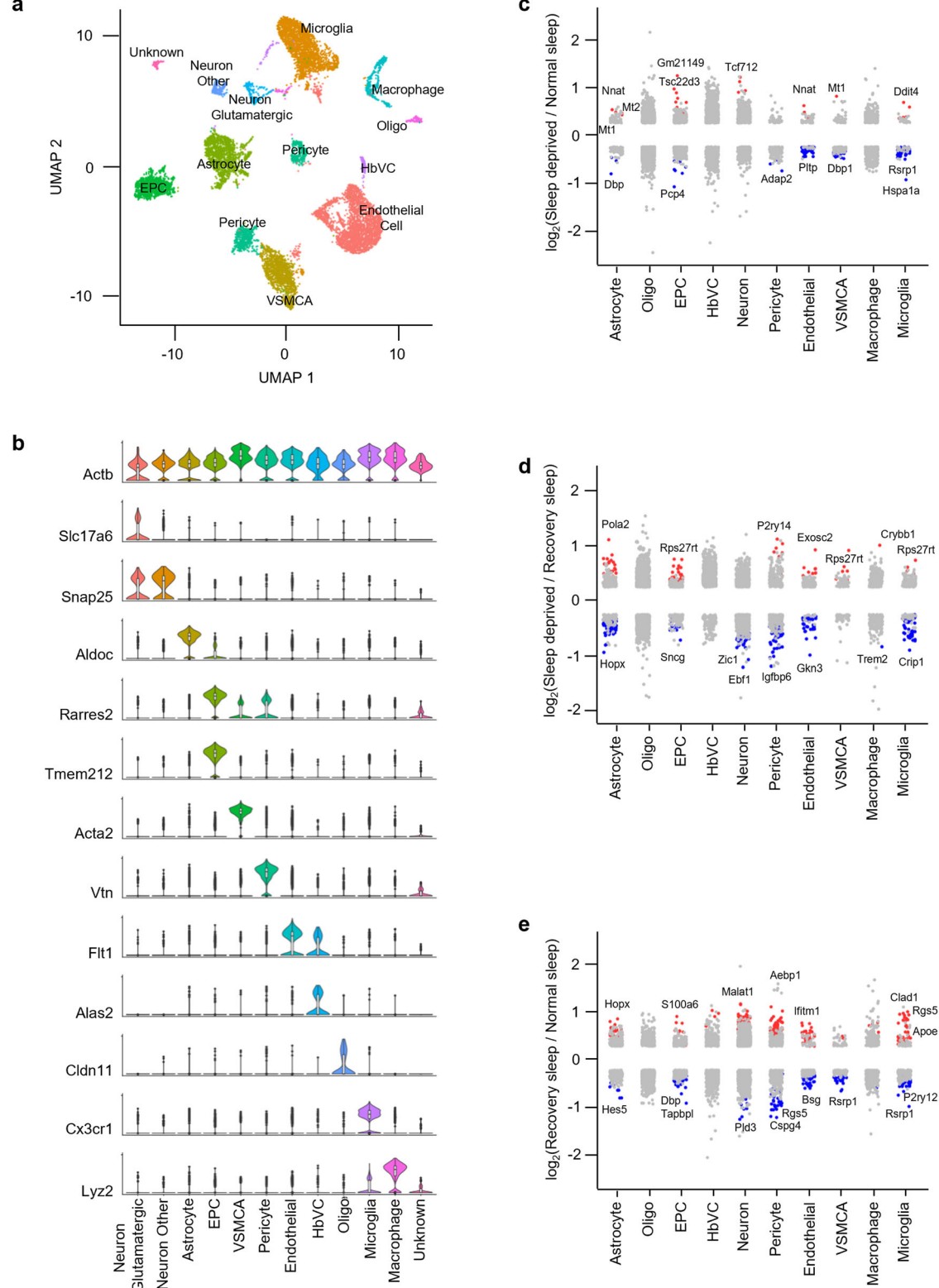

**Fig. 3 Sleep treatments alter the expression profile of cells in the brainstem. a** UMAP visualization of brainstem cells clustered. Cell clusters were colour coded and annotated based on their transcriptional profiles (See details in Methods). **b** Violin plots for each cluster show the expression level of selected known cell-type enriched markers. *Actb* (beta-actin) is shown as a positive control in all cell types (see Supplementary Table 1). **c–e** Strip chart showing changes in gene expression of Sleep Deprived (SD)/Normal Sleep (NS) (**c**), Sleep Deprived (SD)/Recovery Sleep (RS) (**d**), and (**e**) Recovery Sleep (RS)/Normal Sleep (NS) comparisons. Wilcoxon rank-sum tests followed by false discovery rate (FDR) analysis were used to compare the groups. Significantly upregulated and downregulated genes are colour coded with red and blue, respectively (Bonferroni adjusted *P* value < 0.1). Genes in grey are not significantly changed after sleep deprivation (see Supplementary Table 2). EPC ependymocytes, HbVC hemoglobin-expressing vascular cells, Oligo oligodendrocytes, VSMCA Vascular smooth muscle cells, arterial.

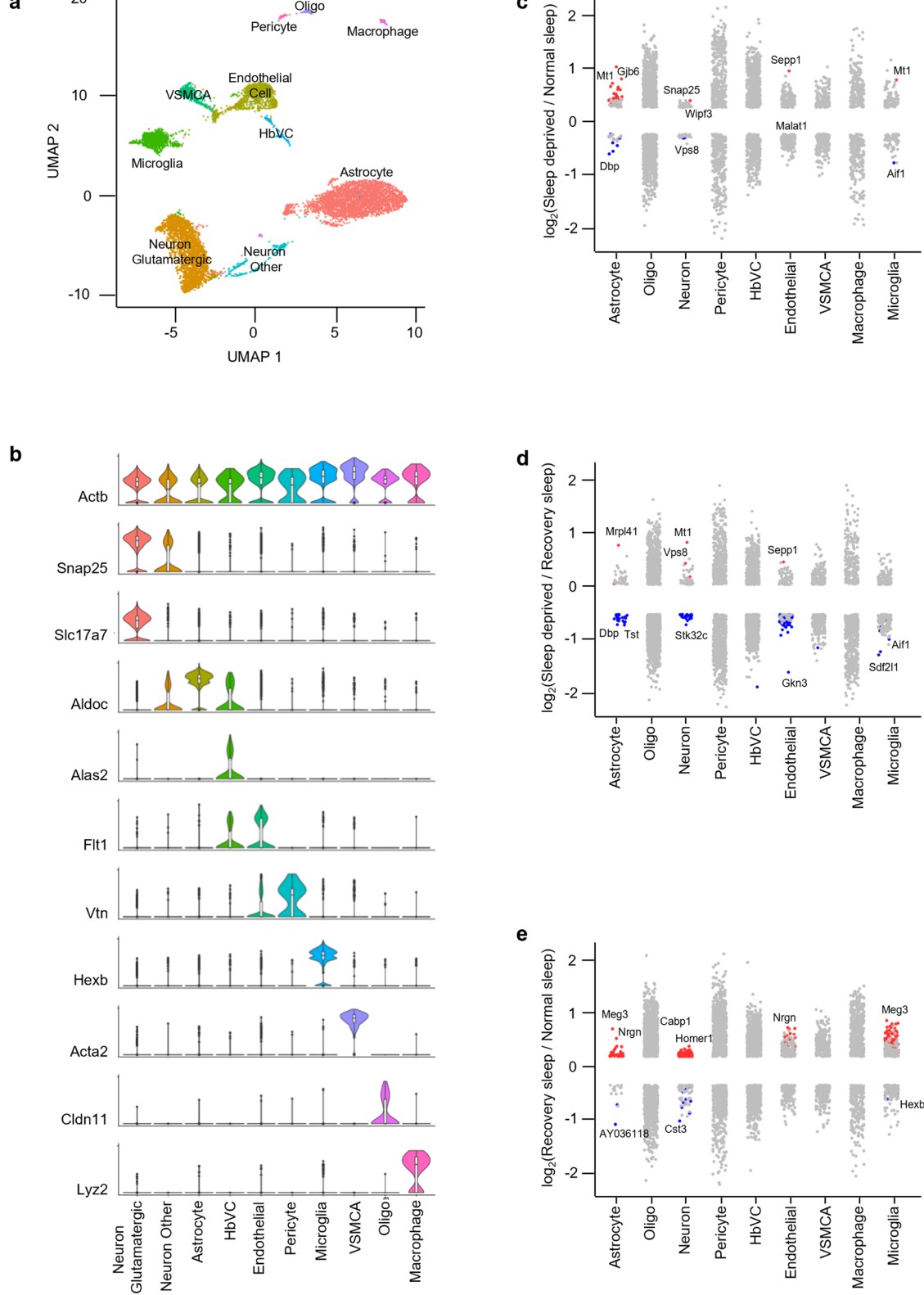

**Fig. 4 Sleep treatments alter the expression profile of cells in the cortex. a** UMAP visualization of cortex cells clustered. Cell clusters were colour coded and annotated based on their transcriptional profiles (See details in Methods). **b** Violin plots for each cluster show the expression level of selected known cell-type enriched markers. *Actb* (beta-actin) is shown as a positive control in all cell types (see Supplementary Table 1). **c–e** Strip chart showing changes in gene expression of Sleep Deprived (SD)/Normal Sleep (NS) (**c**), Sleep Deprived (SD)/Recovery Sleep (RS) (**d**), and (**e**) Recovery Sleep (RS)/Normal Sleep (NS) comparisons. Wilcoxon rank-sum tests followed by false discovery rate (FDR) analysis were used to compare the groups. Significantly upregulated and downregulated genes are colour coded with red and blue, respectively (Bonferroni adjusted *P* value < 0.1). Genes in grey are not significantly changed after sleep deprivation (see Supplementary Table 2). HbVC hemoglobin-expressing vascular cells, Oligo oligodendrocytes, VSMCA Vascular smooth muscle cells, arterial.

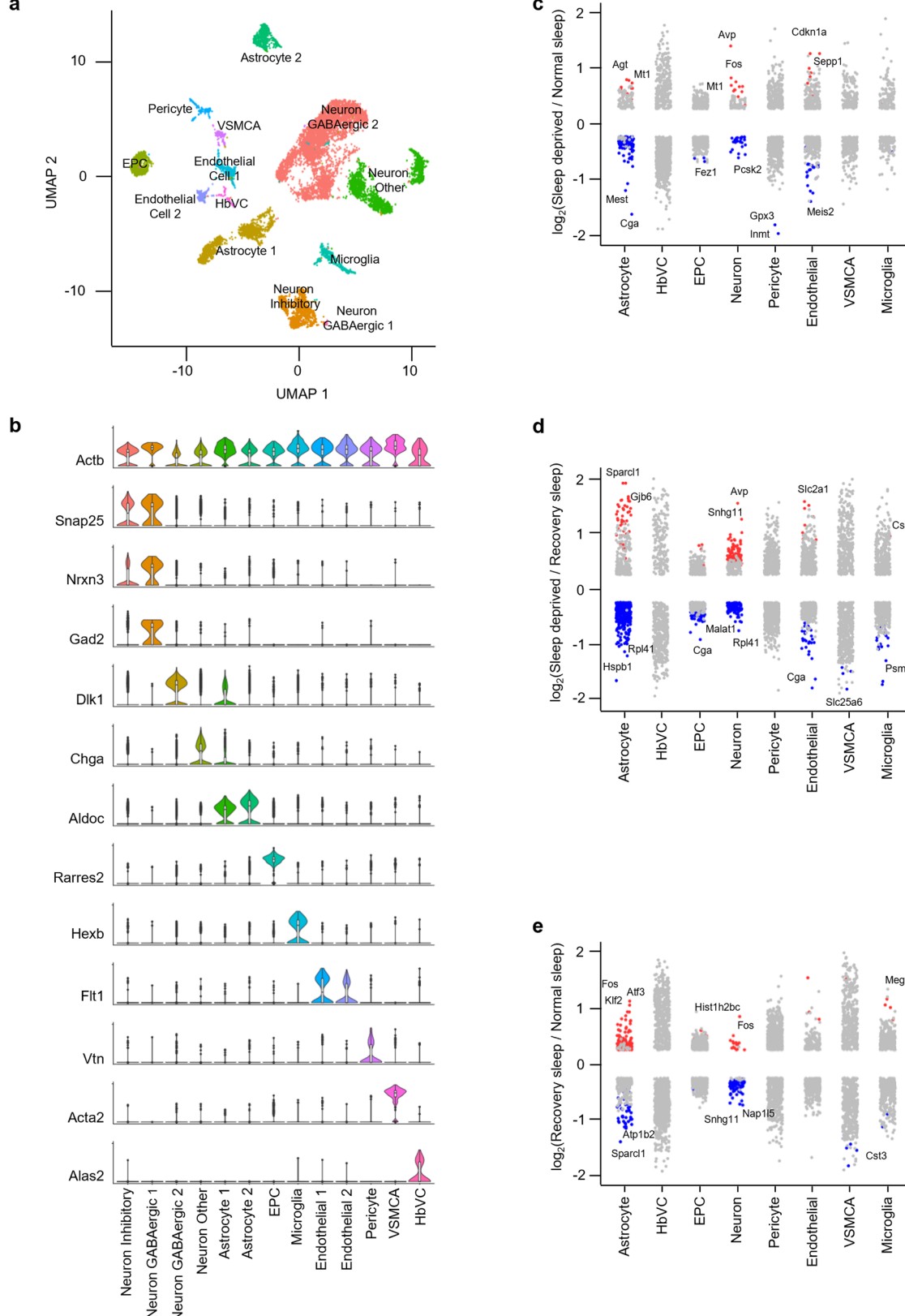

Together, these results show that distinct TFs in each brain region appear to regulate single-cell gene expression.

**Sleep need regulates translational responses of cortical astrocytes and neurons**. Multiple studies in several biological domains have shown that there is a poor relationship between gene and protein expression in global analyses[43–48]. Moreover, in the context of sleep, no studies have been performed to investigate sleep need regulation within specific cell types for proteome shifts in the cortex (or any other brain region). Given that cortex has two principal cell types that are likely to be key in sleep regulation

**Fig. 5 Sleep treatments alter the expression profile of cells in the hypothalamus. a** UMAP visualization of hypothalamus cells clustered. Cell clusters were colour coded and annotated based on their transcriptional profiles (See details in Methods). **b** Violin plots for each cluster show the expression level of selected known cell-type enriched markers. *Actb* (beta-actin) is shown as a positive control in all cell types (see Supplementary Table 1). **c–e** Strip chart showing changes in gene expression of Sleep Deprived (SD)/Normal Sleep (NS) (**c**), Sleep Deprived (SD)/Recovery Sleep (RS) (**d**), and (**e**) Recovery Sleep (RS)/Normal Sleep (NS) comparisons. Wilcoxon rank-sum tests followed by false discovery rate (FDR) analysis were used to compare the groups. Significantly upregulated and downregulated genes are colour coded with red and blue, respectively (Bonferroni adjusted *P* value < 0.1). Genes in grey are not significantly changed after sleep deprivation (see Supplementary Table 2). EPC ependymocytes, HbVC hemoglobin-expressing vascular cells, VSMCA Vascular smooth muscle cells, arterial.

(astrocytes and neurons), we next examined protein expression in these cells. We purified astrocytes and neurons from the cortex using magnetic cell separation (Fig. 8a, Methods) and then performed quantitative global- and phospho-proteomics by using multiplex tandem mass tag (TMT) labeling coupled with liquid chromatography-tandem mass spectrometry (LC-MS/MS) (Fig. 8a, Methods).

We quantified 2454 proteins expressed in astrocytes and 1401 in neurons. To understand how sleep-wake states affect protein abundance in astrocytes and neurons, we compared protein expression in both cell types exposed to different sleep treatments (Fig. 8b and Supplementary Fig. 7a, c). The proportion of altered proteins was higher in neurons (318/1401, 22.7%; 1.5-fold cutoff, FDR < 0.1) than in astrocytes (93/2454, 3.7%; 1.5-fold cutoff, FDR < 0.1) following sleep treatments. We reported similar trends when we compared mRNA expression in cortical astrocytes and neurons (Supplementary Table 2). We hypothesized that this similarity might be a manifestation of astrocyte-neuron communication to maintain synaptic and metabolic homeostasis, as astrocytes bolster the structure and function of neurons. To test this, we performed gene ontology (GO) analysis on differentially expressed proteins in both cell types. In astrocytes, we found prominent alterations of actin filament organization, regulation of cellular component size, synapse organization, and regulation of synapse structure or activity. In contrast, vesicle-mediated transport in synapse, regulation of neurotransmitter levels, and signal release from synapse were enriched in neurons (Fig. 8d, e). Next, we determined these differences are due to sleep treatments, not due to a global difference in protein abundance between the cell types, for this we analyzed the gene annotations of all proteins expressed versus differentially expressed proteins within these cell types (Supplementary Tables 6 and 7).

We also analyzed protein-protein interaction (PPI) networks using STRING to identify biological pathways altered by sleep treatments in astrocytes and neurons[49] (Fig. 8f, g). Sleep-regulated astrocyte proteins segregated into clusters enriched for the spliceosome, fatty acid degradation, and endocytosis (Fig. 8f). Neurons, however, were focused on various networks including the synaptic vesicle cycle, glucose metabolism (glycolysis, gluconeogenesis, TCA cycle), and branched-chain amino acid (valine, leucine, isoleucine) degradation (Fig. 8g). Thus, neurons display prominent responses of vesicular machinery and energy production, in contrast, to support function responses of astrocytes. Interestingly, proteins associated with the endoplasmic reticulum and regulation of actin cytoskeleton function were enriched in both cell types, suggesting a concerted role in sleep.

**Sleep need regulates post-translational responses of cortical astrocytes and neurons**. Protein phosphorylation modulates protein function in a reversible manner[50]. Recent evidence suggests that sleep need modulates protein phosphorylation cycles[51,52], but cell type contributions to these are unknown. We, therefore, performed phosphoproteomics on neuron and astrocyte lysates (Fig. 8a, Methods) and compared protein phosphorylation profiles of sleep treatment groups in both cell types

(Fig. 8c and Supplementary Fig. 7b, d). We identified 1591 phosphorylation sites in astrocytes and 1159 in neurons. Sleep treatments altered 134 phosphorylation sites in astrocytes and 263 in neurons (Fig. 8c, Supplementary Fig. 7b, d). These phosphorylation sites were from 92 and 156 unique proteins from astrocytes and neurons, respectively.

Next, we determined the overlap between sleep-related changes in protein phosphorylation from whole brain proteome data from Wang et al. (2018) and our cortical astrocyte and neuron data and were able to assign cell types to some of the differentially expressed phosphosites found previously[51] (Supplementary Fig. 8a, and Supplementary Table 8). Comparative GO analysis between total versus differentially altered phosphorylated proteins revealed that sleep treatments alter phosphorylation of proteins associated with the regulation of cellular component size, protein complex assembly, and neurotransmitter secretion in astrocytes. By contrast, phosphorylation affects the synaptic vesicle cycle, vesicle-mediated transport in synapses, and calcium-mediated signaling proteins in neurons (Supplementary Tables 9 and 10). Thus, like single-cell transcriptomics, global proteomics and phosphoproteomics demonstrate that sleep need regulates cellular functions in astrocytes and neurons in a cell-specific manner.

## Discussion

In this study, we provide a comprehensive single-cell transcriptome profiling of the cell types within three brain areas (brainstem, cortex, and hypothalamus) of mice in different sleep stages. We found sleep need modulates transcriptional patterning of astrocytes, neurons, endothelial cells, and microglia in all three brain regions. We also utilized our single-cell gene expression information to interrogate remodeling of astrocyte-neuron signaling during sleep and determined expression changes in transcription factors that may mediate these effects. Cell-specific proteomics reveals that sleep need regulate translational and post-translational responses of astrocytes and neurons in the cortex differently.

Previous bulk RNA quantification methods could not reveal the cell-specific differential expression changes that we found in this study. Our differential gene expression analysis within individual cell types did not show a universal sleep-related changes, but rather a distinct pattern of transcriptional responses of sleep need in each cell type. Moreover, this was not the same for different brain regions, further highlighting those cells in their local milieu respond to the same stimulus (sleep deprivation and recovery) in different ways. The transcriptional responses of sleep need also vary between the cortex and hypothalamus when RNA is quantified in bulk RNA-seq[28–30]. Indeed, in our analyses, we were able to retrospectively assign gene expression changes seen in these bulk microarray or sequencing studies to specific cell types, enhancing the utility of these studies further.

Single-cell RNA-seq analysis has also advanced our understanding of the functional pathways affected by sleep perturbations. Pathway analysis revealed different patterns of sleep-driven changes across cell types and brain areas. For example, we found that sleep need alters gene expression associated with ribosomal

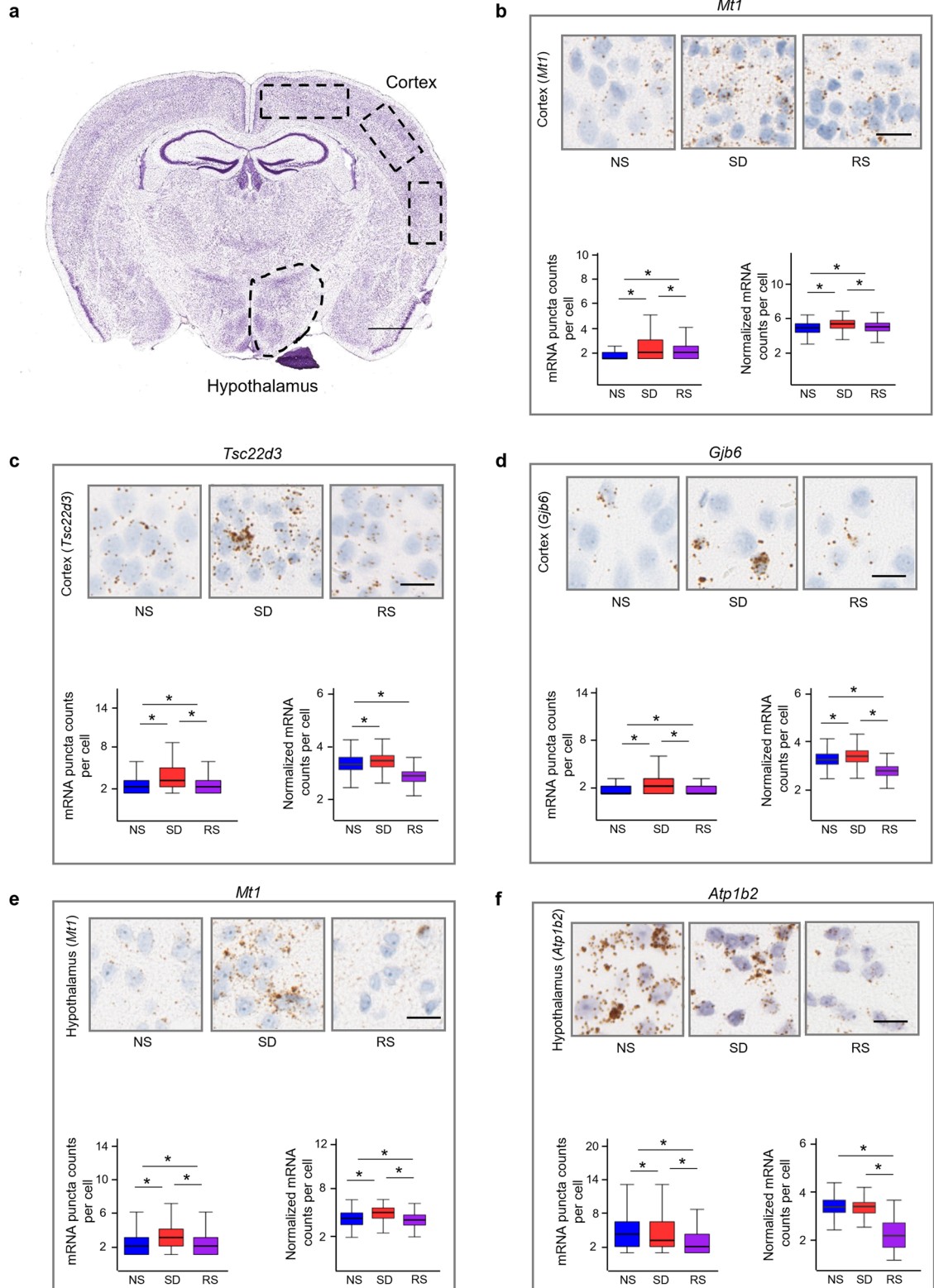

**Fig. 6 In situ hybridisation validation of transcriptional changes following sleep treatments. a** Regions of interests (ROIs) for RNAscope analysis are shown on the coronal micrograph, scale bar, 1 mm (from Allen Brain Atlas). **b–f** Comparison of RNAscope single cell in situ hybridization and sequencing of (**b**) *Mt1*, (**c**) *Tsc22d3*, (**d**) *Gjb6* expression in cortex, and (**e**) *Mt1*, (**f**) *Atp1b2* expression in hypothalamus across sleep treatment groups. Micrographs of mouse cortices and hypothalamic shown. mRNA molecules are brown dots as revealed by Diaminobenzidine (DAB), and nuclei are counterstained blue (with haematoxylin). Scale bar, 20 μm. mRNA were counted from cells of $n = 3$ biological replicate brains per group. Kruskal–Wallis test followed by post-hoc Dunn test. Boxplots represent lower bases as minimum, upper top as maximum, and dark horizontal lines within boxes as median. *$P < 0.01$. NS normal sleep, SD sleep deprived, RS recovery sleep. For sample sizes and statistics, see Supplementary Data 1.

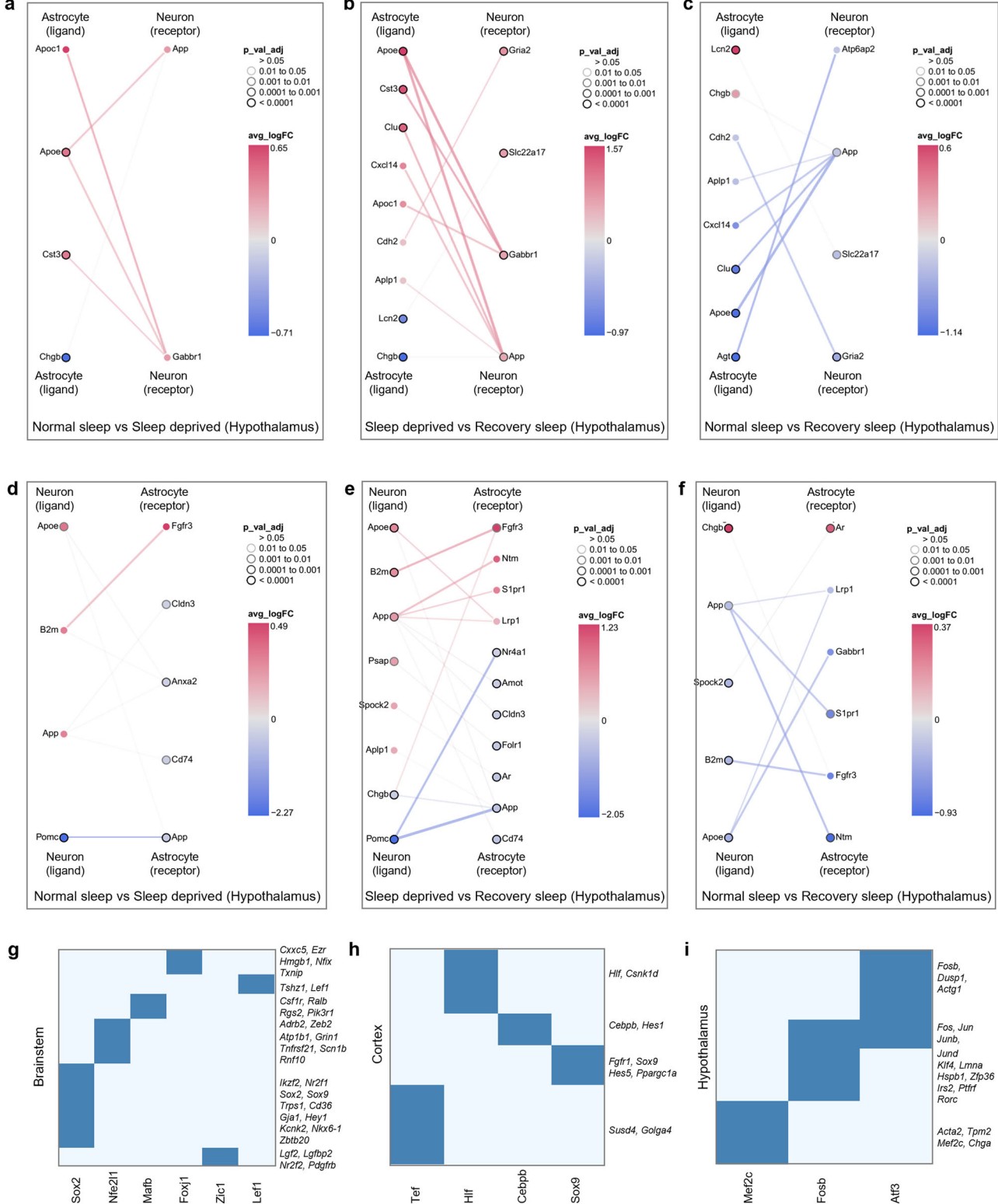

**Fig. 7 Sleep need modulates cell-cell communication and gene regulatory networks. a–f** Interactions between differentially expressed ligands of astrocytes and receptors of neurons for (**a**) NS versus SD, (**b**) SD versus RS, and (**c**) NS versus RS comparisons, and ligands of neurons and receptors of astrocytes for (**d**) NS versus SD, (**e**) SD versus RS, and (**f**) NS versus RS comparisons in the hypothalamus. Nodes represent ligands or receptors in astrocytes and neurons, respectively. Node outline thickness indicates the level of significance (Benjamini-Hochberg adjusted P-value). Colour of nodes and edges represents the magnitude of alteration in expression (log fold-change, logFC) (see Methods). **g–i** Uniquely expressed transcription factors (rows) in sleep-deprived cells and their co-expressed gene sets (columns) in (**g**) brainstem, (**h**) cortex, and (**i**) hypothalamus. For specific GO terms of the co-expressed genes, see Supplementary Table 5. NS normal sleep, SD sleep deprived, RS recovery sleep.

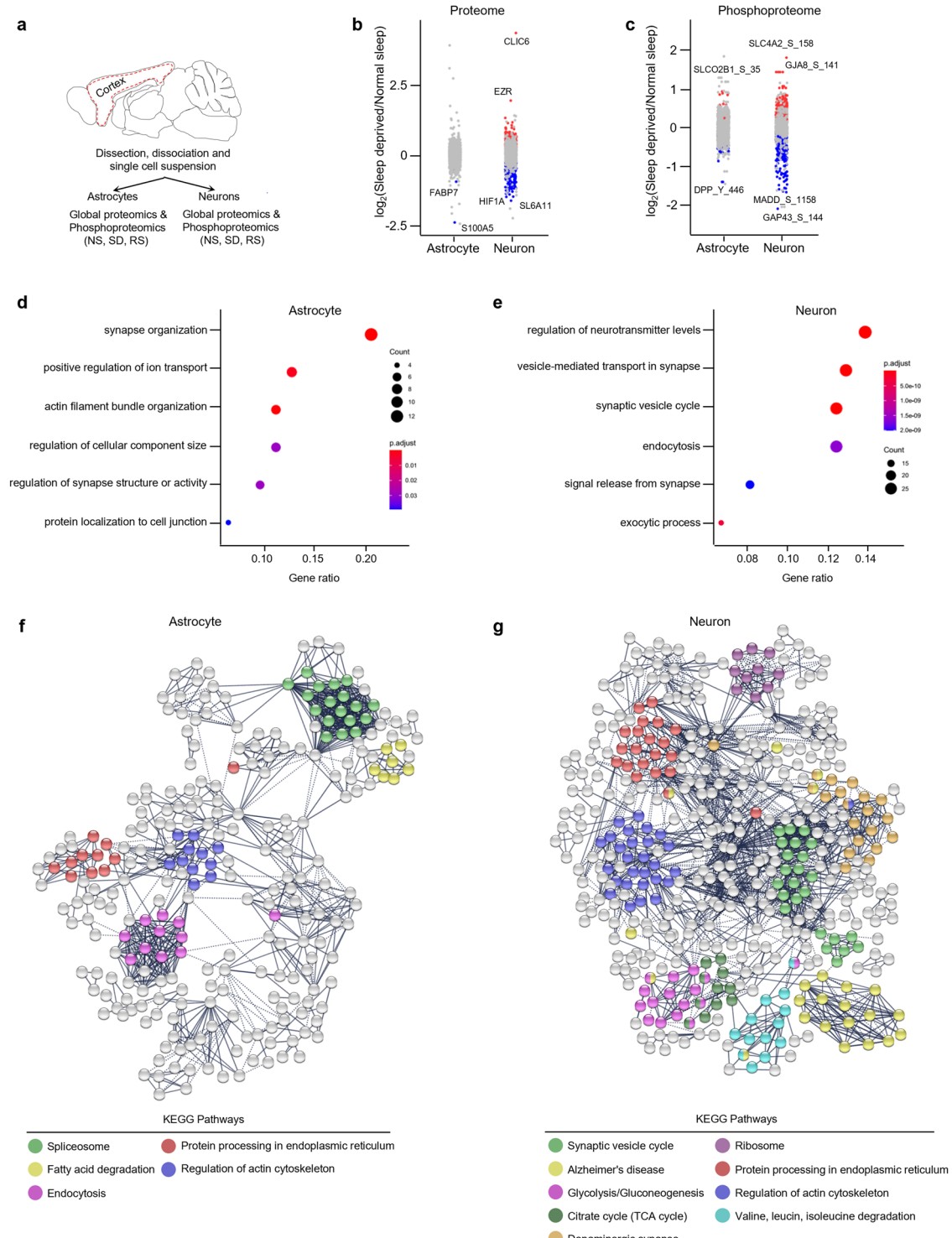

**Fig. 8 Sleep deprivation regulates the global and phosphoproteome of astrocytes and neurons in the cerebral cortex. a** Schematic of proteomics workflow. Dotted red line indicates the area of the cortex dissected for magnetic cell separation (see Methods). **b, c** Strip charts showing changes in (**b**) global and (**c**) phosphoproteome for Sleep Deprived (SD)/Normal Sleep (NS) comparisons in astrocytes and neurons. Unpaired t-tests followed by false discovery rate (FDR) analysis were used to compare groups. Significantly upregulated and downregulated protein expression is colour coded with red and blue, respectively (Benjamini-Hochberg adjusted $P$ value < 0.1). Proteins in grey are not significantly changed after sleep deprivation. **d, e** Dot plots showing significantly enriched top functional annotations from global proteomes of (**d**) astrocytes and (**e**) neurons. Circle size is proportional to protein count and level of significance (Benjamini-Hochberg adjusted $P$ value < 0.05) is colour coded (see Supplementary Tables 6 and 7). **f, g** Interaction networks of significantly altered (FDR < 0.1) global proteins in (**f**) astrocytes and (**g**) neurons across sleep treatments. Functional enrichment of proteins (KEGG pathways) in the network is colour coded. The networks were generated using STRING Version 11.0 database (PPI enrichment $P$ value < 0.001).

biogenesis in astrocytes, but vesicle-mediated transport in synapses and nucleotide/nucleoside phosphate metabolism in neurons. Endothelial cells exhibit changes in protein folding pathways, whereas microglia show neuroinflammatory responses. Interestingly, protein folding and neuroinflammatory responses have been documented in previous sleep studies[17,31,32,53]. To the best of our knowledge, these results provide cell specificity to these sleep-related responses for the first time. Furthermore, the same cell type in different brain areas performed variable molecular functions to regulate diverse biological processes, indicating brain region-specific molecular responses of sleep need. Collectively, these data indicate that the transcriptional response of sleep loss is not identical in all cell types and brain areas, rather this response is cell type, and brain region-specific.

How are the specific transcriptional changes brought about? We were able to identify transcription factors (TF) that mediate gene expression changes in each cell type in sleep-deprived cells from the brainstem, cortex, and hypothalamus. Interestingly, we found no common TF in any two brain areas. This analysis revealed that the key TFs in the brainstem are Sox2, Nfe2l1, Mafb, Foxj1, Zic1, and Lef1, whereas in the cortex they were Tef1, Hlf1, Cebpb, and Sox9, and in hypothalamus Mef2c, Fosb and Atf3. Future studies will explore directly (e.g., using loss-of-function approaches) the roles of these TFs in sleep homeostasis.

Astrocytes and neurons are major cell types of the brain that are likely to be important in sleep. Since we were able to delineate transcriptional changes in each of these cell types, this enabled us to determine new relationships between the two cell types in sleep. We generated networks of receptor-ligand interactions to decipher sleep-driven astrocyte-neuron communications in all three brain regions. The significance of these interactive networks is of high importance as some recent findings show that astrocytes interact with neurons to modulate sleep circuitry, and metabolic coupling between them, across the sleep-wake cycle[35,36,54]. Thus, the discovery of these factors, and their source and target, is a fruitful avenue to define therapeutic interventions of sleep abnormalities. We found Apoe, Pomc and Bdnf signaling were modulated by sleep in the different brain regions. Of note, the role of these genes in sleep-related pathologies have been described[37,39,55]. Our study advances our understanding by showing the cell types involved, and the directionality of changes in intercellular ligand-receptor signaling. Such insights will enable further mechanistic studies targeting these ligands and receptors to modulate sleep, and potentially develop therapeutic agents that manipulate these pathways.

To gain further insights into the functional implication of sleep need in astrocytes and neurons, we performed proteomics and phosphoproteomics. We found that 22% of neuronal proteins responded to sleep perturbation compared to only 3% in astrocytes. This suggests that astrocytes are less responsive to acute sleep pressure than neurons. Interestingly, a previous sleep study using a 6 h sleep deprivation paradigm concluded that the whole brain proteome remains globally stable following sleep treatments[51]. There are some possible explanations of the disparity between those findings and ours. First, differences could be due to reciprocal changes in different brain areas and/or cell types that may have been "masked" by bulk cell proteomics, particularly in whole brain samples; our study used cortex only. Second, 6 h of sleep deprivation (compared to 12 h used here) may not have been sufficient time for transcriptional changes to become apparent at the protein level. We propose that astrocytes show relative resistance to sleep need and may thereby provide stable support to rapidly changing neurons via metabolic coupling[36]. The biological functions of sleep-mediated alterations include actin filament bundle organization, positive regulation of ion transport, synapse organization, and fatty acid degradation in astrocytes, and vesicle-mediated transport in synapse, regulation of neurotransmitter levels, and glucose metabolism in neurons. These functions support the proposed importance of synapse and energy homoeostasis as functions of sleep-wake cycles[5,6]. Our results now provide cell-specific compartmentalization of these functions.

Sleep need influences the circadian pacemaker. However, the molecular mechanism of this interaction is largely unknown[9]. Sleep deprivation does not affect clock gene expression within the mammalian central pacemaker, the suprachiasmatic nucleus (SCN), but does modulate extra-SCN clock gene expression[40,56,57]. This indicates sleep need could influence daily gene expression rhythms in brain areas outside of the SCN. Sleep deprivation completely abrogates the daily oscillation of forebrain synaptic proteins and their phosphorylation[15,52]. This is not confined to the brain, as Lu and colleagues reported the loss of daily rhythms of 87% of oscillating transcripts within the lungs in response to sleep loss. Interestingly, clock gene expression in the lung remained rhythmic after sleep deprivation[58]. Although our study focuses on the unimodal changes of the transcription, translation, and post-translation at the one time of day (ZT12), information about cell-specific alterations in gene and protein expression will be useful to design targeted time course experiments to understand how sleep need affects molecular oscillations in a cell-specific manner.

We reported diverse functional annotations of gene and protein (and phosphoprotein) expression in the cells from all three brain regions. This indicates that sleep is regulated differentially at transcriptional, translational, and post-translational levels within individual cell types. The ensemble of cellular processes regulated at these levels include synapse organization, synaptic vesicle recycling, protein folding, ribosome function, and glucose metabolism. As such, we propose that prolonged wakefulness during sleep deprivation leads not only to sleepiness, but could also impair cognition, physiology, and metabolism via the diverse molecular responses we identified in individual brain cells. Moreover, we suggest that sleep need could be a compensatory mechanism to reverse such molecular changes in the brain. Overall, these results will help to advance a variety of efforts towards understanding the cellular and molecular mechanisms of sleep and wake and provide a critical path to dissect sleep functions.

## Methods

**Animals**. All animal studies were carried out in concordance with an approved protocol from the Institutional Animal Care and Use Committee (IACUC) at Perelman School of Medicine at the University of Pennsylvania or under license by the United Kingdom Home Office under the Animals (Scientific Procedures) Act 1986, with Local Ethical Review by the Francis Crick Institute Animal Welfare & Ethical Review Body Standing Committee (AWERB). Wild type, male C57BL/6 J mice were purchased from Charles River and allowed to acclimatize in the animal unit for at least 2 weeks prior to experiments. Mice were selected such that, at the time of experiments, they were aged between 9–11 weeks. Before sleep deprivation, mice were singly housed in automated sleep fragmentation chambers (Model #80391, Campden/Lafayette Instrument Lafayette, IN, USA) for habituation with *ad libitum* access to food and water under standard humidity and temperature (21 ± 1 °C) on a 12-h light: 12-h dark cycle[59].

**Sleep deprivation**. Sleep deprivations were performed as previously demonstrated and adapted to a newer device model (80391), which enables a much faster sweep time[60]. Briefly, this device acts by applying tactile stimulus with a horizontal bar sweeping just above the cage floor (bedding) from one side to the other side of the cage. Once the sweeper is on, animals need to step over approaching the sweeper to resume their unrestrained behaviors. Sleep deprivation was initiated at lights on [Zeitgeber Time 0 (ZT0)] by switching on the motors, choosing the continuous sweeping mode (approximately 7.5 s cycle time), and stopped after 12 h at lights off (ZT12). In addition to the sweeping bar, additional attempts were made to maintain wakefulness mostly during the second 6 h of sleep deprivation by occasionally tapping on the cage or gentle touches with a brush. We opted for a 12-h long sleep deprivation timeline with the aim to observe the transcriptional changes

in protein abundance, as we performed both transcriptomics and proteomics assays. To assess how 12-h of sleep deprivation affected the sleep/wake behaviors, 3 days baseline EEG/EMG were recorded after mice were acclimated for a week. Mice were recorded for the next 3–4 days (which encompassed sleep deprivation and recovery phases). Post sleep deprivation, animals were allowed to recover for 24-h. Baseline recording 1-day before sleep deprivation was used as the control condition (normal sleep). Following sacrifice by cervical dislocation, whole brains were isolated at ZT12 (i.e. lights off) for *ad libitum* Normal Sleep (NS, ZT0-12), Sleep Deprivation (SD, ZT0-12), and 12-h sleep deprivation followed by 24-h of Recovery Sleep (RS, ZT12-12) groups. Isolated whole brains were placed in ice-cold Hibernate EB (BrainBits LLC, HEB) media, and brainstem, cortex, and hypothalamus were quickly dissected as these areas are involved in the regulation of sleep-wake states[13,14], and tissue collected from three animals per sleep treatment group per brain area pooled for the preparation of single-cell suspensions.

**Sleep phenotyping**. Mice aged between 9–11 weeks were implanted with a telemetry transmitter (HD-X02, Data Sciences International, St. Paul, MN, USA) connected to electrodes for continuous EEG (Electroencephalography)/EMG (Electromyography) recording. Under anaesthesia (isoflurane; induction 3–4%, maintenance 2–2.5%), two stainless steel EEG electrodes (length of screw shaft: 2.4 mm; head diameter: 2.16 mm; shaft diameter: 1.19 mm; Plastics One, Roanoke, VA, USA) were implanted epidurally over the right frontal and parietal cortices, as previously described[61]. The electrodes were connected to the telemetry transmitter via medical-grade stainless steel wires. The EEG electrodes were covered with dental cement (Kemdent, Purton, Swindon, UK). Two EMG stainless-steel leads were inserted into the neck muscles ~5 mm apart and sutured into place. The telemetry transmitter was placed in a subcutaneous pocket and positioned along the left dorsal flank. Analgesia was administered at the onset of the surgery (subcutaneous injection of buprenorphine (Vetergesic) at 0.1 mg/kg and meloxicam (Metacam) at 10 mg/kg). Animals were allowed to recover for a minimum of 10 days before being subjected to experimental protocols. EEG/EMG signals were recorded continuously for 6–7 days using Data Sciences International hardware and Dataquest ART software (Data Sciences International, St. Paul, MN, USA). The EEG/EMG data were transmitted at 455 kHz to an RPC-1 receiver (Data Science International) and sampled at 250 Hz.

**Single-cell suspensions and library preparation**. The protocol for obtaining single-cell suspensions for all three brain regions was adapted from that used by Holt and colleagues[62]. Briefly, tissue dissections were dissociated with Papain (BrainBits LLC, PAP/HE) following the manufacturer's instruction followed by manual trituration using fire-polished Silanized Pasteur pipette and filtration through 70 µm cell strainer (Miltenyi Biotec, 130-095-823). Cells were pelleted at 300 × g, 10 min, the supernatant carefully removed, and cells resuspended in a minimal volume of PBS with 0.5% of BSA (Sigma-Aldrich, A7906). To reduce debris, we incubated the cell suspensions with myelin removal beads (Miltenyi Biotec, 130-096-733) and passed this through a LS Column (Miltenyi Biotec, 130-042-401) with 70 µm pre-separation filter. Flow-through contained unlabelled cells, which were collected and counted (Countess II Automated Cell Counter; Cat # AMQAX1000) estimate yield and viability. Library preparation was carried out with a 10X Genomics Chromium Single Cell Kit Version 2. Suspensions were prepared as described above and diluted in PBS with 0.5% of BSA to concentration ~350 cells/ µl and added to 10X Chromium RT mix to achieve a loading target between 7000–12,000 cells. After cell capture, downstream cDNA synthesis, library preparation, and sequencing were performed according to the manufacturer's instructions (10X Genomics).

**Astrocyte and neuron separation from cortical single cell suspension**. To perform cell type-specific proteomics and phosphoproteomics, astrocytes and neurons were separated from cortical single-cell suspensions as previously described[62] with slight modifications. Single-cell suspensions were incubated with FcR Blocking Reagent (Miltenyi Biotec, 130-092-575) and Anti-ACSA-2 MicroBeads (Miltenyi Biotec, 130-097-678) for 10–15 min at 2–8 °C. Cells were spun down for 10 min at 300 × g at room temperature and resuspended in a minimal volume of PBS with 0.5% of BSA before passing through a LS Column. Flow-through was collected for further neuron separation, the LS Column that retained astrocytes was removed from the magnetic field and the ACSA-2 labeled astrocytes were eluted. Cells were pelleted from flow-through at 300 × g, 8 min, the supernatant carefully removed, and resuspended in a minimal volume of PBS with 0.5% of BSA. Suspensions were incubated with Non-Neuronal Cell Biotin-Ab cocktail (Miltenyi Biotec, 130-115-389) for 5 min and Anti-Biotin MicroBeads (Miltenyi Biotec, 130-090-485) for 10 min at 2–8 °C, respectively. Following incubation, suspensions were passed through a LS Column placed in a magnetic field and non-labeled neuronal cells were collected in flow-through.

**TMT-based quantitative proteomics and phosphoproteomics**. Cell lysis and protein extraction were performed in the presence of protease and phosphatase inhibitors. Protein precipitation was performed with 1:6 volume of pre-chilled (−20 °C) acetone for overnight at 4 °C. After overnight incubation, lysates were centrifuged at 14,000 g for 15 min at 4 °C. Supernatants were discarded without

disturbing pellets, and pellets were air-dried for 2–3 min to remove residual acetone. Then pellets were dissolved in 300 µL 100 mM TEAB buffer. Sample processing for TMT-based quantitative proteomics and phosphoproteomics was performed following the same protocol as described previously[63]. In brief, protein concentration in each sample was determined by using the Pierce™ BCA Protein Assay Kit (Thermo Fisher Scientific, 23225). 75 µg protein per condition was transferred into new microcentrifuge tubes and 5 µL of the 200 mM TCEP was added to reduce the cysteine residues and the samples were then incubated at 55 °C for 1h. Subsequently, the reduced proteins were alkylated with 375 mM iodoacetamide (freshly prepared in 100 mM TEAB) for 30 min in the dark at room temperature. Then, trypsin (Trypsin Gold, Mass Spectrometry Grade; Promega, V5280) was added at a 1:40 (trypsin: protein) ratio and samples were incubated at 37 °C for 12-h for proteolytic digestion. After in-solution digestion, peptide samples were labelled with 10-plex TMT Isobaric Label Reagents (Thermo Fisher Scientific, 90113) following the manufacturer's instructions. The reactions were quenched using 5 µL of 5% hydroxylamine for 30 min. Proteins from astrocytes or neurons for the three experimental conditions, i.e. NS, SD, and RS (3 biological replicates for each) were labeled with the nine TMT reagents within a TMT 10-plex reagent set, while the 10th reagent was used for labeling an internal pool containing an equal amount of proteins from each sample. Application of multiplexed TMT reagents allowed comparison of NS, SD, and RS samples within the same MS run, which eliminated the possibility of run-to-run (or batch) variations.

In order to perform phosphoproteome analysis, TMT labelled peptides were subjected to TiO$_2$-based phosphopeptide enrichment according to the manufacturer's instructions (High-Select™ TiO$_2$ Phosphopeptide Enrichment Kit, Thermo Fisher Scientific, A32993). The flow-through and washes from the TiO$_2$ enrichment step were combined, evaporated to dryness and subjected to Fe-NTA-based phosphopeptide enrichment according to the manufacturer's instructions (High-Select™ Fe-NTA Phosphopeptide Enrichment Kit, Thermo Fisher Scientific, A32992). Phosphopeptides enriched using TiO$_2$- and Fe-NTA-based methods were combined, evaporated to dryness.

TMT-labelled samples (both for global and phosphoproteomics analysis) was resuspended in 5% formic acid and then desalted using a SepPak cartridge according to the manufacturer's instructions (Waters, Milford, Massachusetts, USA). Eluate from the SepPak cartridge was evaporated to dryness and resuspended in buffer A (20 mM ammonium hydroxide, pH 10) prior to fractionation by high pH reversed-phase (RP) chromatography using an Ultimate 3000 liquid chromatography system (Thermo Scientific). In brief, the sample was loaded onto an XBridge BEH C18 Column (130 Å, 3.5 µm, 2.1 mm × 150 mm, Waters, UK) in buffer A and peptides eluted with an increasing gradient of buffer B (20 mM Ammonium Hydroxide in acetonitrile, pH 10) from 0–95% over 60 min. The resulting fractions (8 fractions per sample) were evaporated to dryness and resuspended in 1% formic acid prior to analysis by nano-LC MSMS using an Orbitrap Fusion Lumos mass spectrometer (Thermo Scientific).

**Nano-LC Mass Spectrometry**. High pH RP fractions were further fractionated using an Ultimate 3000 nano-LC system in line with an Orbitrap Fusion Lumos mass spectrometer (Thermo Scientific). In brief, peptides in 1% (vol/vol) formic acid were injected onto an Acclaim PepMap C18 nano-trap column (Thermo Scientific). After washing with 0.5% (vol/vol) acetonitrile 0.1% (vol/vol) formic acid peptides were resolved on a 250 mm × 75 µm Acclaim PepMap C18 reverse phase analytical column (Thermo Scientific) over a 150 min organic gradient, using 7 gradient segments (1–6% solvent B over 1 min, 6–15% B over 58 min, 15–32%B over 58 min, 32–40%B over 5 min, 40–90%B over 1 min, held at 90%B for 6 min and then reduced to 1%B over 1 min) with a flow rate of 300nL min$^{-1}$. Solvent A was 0.1% formic acid and Solvent B was aqueous 80% acetonitrile in 0.1% formic acid. Peptides were ionized by nano-electrospray ionization at 2.0 kV using a stainless-steel emitter with an internal diameter of 30 µm (Thermo Scientific) and a capillary temperature of 275 °C. All spectra were acquired using an Orbitrap Fusion Tribrid mass spectrometer controlled by Xcalibur 4.1 software (Thermo Scientific) and operated in data-dependent acquisition mode using an SPS-MS3 workflow. FTMS1 spectra were collected at a resolution of 120,000, with an automatic gain control (AGC) target of 200,000 and a max injection time of 50 ms. Precursors were filtered with an intensity threshold of 5000, according to charge state (to include charge states 2–7) and with monoisotopic peak determination set to Peptide. Previously interrogated precursors were excluded using a dynamic window (60 s +/−10ppm). The MS2 precursors were isolated with a quadrupole isolation window of 0.7 m/z. ITMS2 spectra were collected with an AGC target of 10,000, max injection time of 70 ms and CID collision energy of 35%. For FTMS3 analysis, the Orbitrap was operated at 50,000 resolutions with an AGC target of 50,000 and a max injection time of 105 ms. Precursors were fragmented by high energy collision dissociation (HCD) at a normalized collision energy of 60% to ensure maximal TMT reporter ion yield. Synchronous Precursor Selection (SPS) was enabled to include up to 5 MS2 fragment ions in the FTMS3 scan.

**RNAscope in situ hybridization**. Brains were quickly isolated and fixed in 10% neutral-buffered formalin (NBF) for 24-h, followed by ethanol gradient dehydration and infiltration with melted paraffin in an automated processor. An array of 4 µm coronal sections of whole-brain was prepared, to collect different regions of the cortex, and RNAscope in situ hybridization was performed according to the

manufacturer's reference guide, using the single-plex RNAscope® assay kit-BROWN (Advanced Cell Diagnostics) on a Leica Biosystems BOND RX platform, as described previously[64]. Brain slices were baked and deparaffinized on the instrument, followed by target retrieval and protease treatment. Hybridization was performed by using probe *Mt1* (ACD, 547711), *Tsc22d3* (ACD, 448341), *Gjb6* (ACD, 458811) and *Atp1b2* (ACD, 417131) followed by amplification as these genes were among our top hits (based on FDR and fold changes), DAB chromogenic detection and counterstain with haematoxylin. Imaging of sections was carried out on a Leica Biosystems Aperio AT2 Digital Pathology Scanner and analysis performed using Visiopharm software.

**Sleep scoring and spectral analysis**. Vigilance states for consecutive 4-sec epochs were classified by visual inspection of the EEG and EMG signals, according to standard criteria, as follows: wakefulness (high and variable EMG activity and a low amplitude EEG signal), non-rapid eye movement sleep (NREMS; high EEG amplitude, dominated by slow waves and low amplitude EMG), and rapid eye movement sleep (REMS; low EEG amplitude, theta oscillations of 5–9 Hz, and loss of EMG muscle tone). EEG power spectra were computed for consecutive 4-sec epochs using Welch's method in Matlab 2017a (MathWorks, Natick, MA, USA; frequency range, 0.25–25 Hz; resolution, 0.25 Hz; Hanning window function). Epochs containing EEG artefacts were discarded from EEG spectral analyses (% of recording time: $7.9 \pm 0.8\%$). EEG power spectra were determined for NREMS, REMS, and wakefulness during the 12-h light-dark periods of the 3 recording days. EEG power spectra are expressed as a percentage of total EEG power (frequency range: 1–25 Hz; resolution: 1 Hz). EEG delta power total and during NREMS were computed by adding the EEG power in the frequencies ranging from 1 to 4.5 Hz. Averaged over consecutive intervals to which an equal number of 4-s NREM sleep epochs contribute, and then expressed as a percentage of levels reached between ZT8-12 during the day 1 (baseline).

**Analysis of single-cell RNA-seq data**. FASTQ files containing sequence reads were mapped to the mouse reference genome GRCm38 using Cell Ranger Version 2.2.0 on a Linux high-performance computing system. The output of the Cell Ranger pipeline (filtered counts) was parsed into R version 4.0.0 to perform downstream analysis with the various R packages – Seurat version 3.1.5[65], Monocle version 2.10.1[66] and SCENIC version 1.1.2.2[38]. The data were first normalized and log-transformed using Seurat function NormalizeData. Differential gene expression was computed by Wilcoxon rank sum test method available in the Seurat function FindMarkers (recommended). Cluster analysis on the first 13 principal components was performed after calculating the JackStraw. Clusters were visualized with uniform manifold approximation and projection (UMAP) in Seurat with default settings. Seurat functions UMAPPlot, VlnPlot, FeaturePlot, DotPlot, and DoHeatmap were adapted for visualization of gene expression. All Monocle and SCENIC analyses were performed on a Linux computing system using 256 GB RAM spread across 32 compute cores. The expression matrix from the Seurat data object was then used to perform Single Cell regulatory Network Inference and Clustering analysis with SCENIC. The species-specific RcisTarget database of motif rankings were downloaded from https://resources.aertslab.org/cistarget/databases/mus_musculus/mm10/refseq_r80/mc9nr/. The standard SCENIC workflow was run on each dataset individually or on a merged data together as described at http://scenic.aertslab.org.

**Determination of cell identity clusters**. To determine the cell-type identity of each cluster from the brainstem, cortex, and hypothalamus, we used multiple cell-type-specific/enriched markers genes that have been previously described in the single-cell transcriptomics of the mouse brain in the literature[24–27]. Each cluster showing the high expression level of known markers specific to a particular cell was considered as a cluster of that cell type (Supplementary Table 1). In general, we selected the threshold of average fold change above 2-3 and false discovery rate (FDR) <0.001. We also cross-validated the other clusters from the same brain area for the absence of these markers. We found the available markers were sufficient to define all major cell types in all three brain regions, with one "unknown" (unclassifiable) cell type in the brainstem.

**Astrocyte-neuron communication networks**. Astrocyte-neuron communication networks were predicted by previously described intercellular communication networks[67,68]. The cell interactomes for differentially expressed genes were created based on ligand-receptor interactions. In the networks, nodes represent ligands and receptors expressed in astrocytes and neurons and, vice-versa. The border of nodes represents the level of significance (FDR). Edges represent interactions between them, and the color of nodes and edges represents the magnitude of differential expression.

**Database search and statistical analysis of quantitative proteomics data**. Quantitative proteomics and phosphoproteomics raw data files were analysed using the MaxQuant computational platform (*version* 1.5.2.8) with the Andromeda search engine[69]. MS2/MS3 spectra were searched against UniProt database specifying *Mus musculus* (Mouse) taxonomy (Proteome ID: UP000000589; Organism ID: 10090; Protein count: 52026). All searches were performed using "Reporter ion MS3" with "10-plex TMT" as isobaric labels with a static modification for cysteine alkylation (carbamidomethylation), and oxidation of methionine (M) and protein N-terminal acetylation as the variable modifications. Phospho (STY) was included as an additional variable modification for the phosphopeptide enrichment analyses. Trypsin digestion with maximum two missed cleavages, minimum peptide length as seven amino acids, precursor ion mass tolerance of 5 ppm and fragment mass tolerance of 0.02 Da were specified in all analyses. The false discovery rate (FDR) was specified at 0.01 or 0.05 for peptide spectrum match (PSM), protein and site decoy fraction. TMT signals were corrected for isotope impurities based on the manufacturer's instructions. Subsequent processing and statistical analysis of quantitative proteomics and phosphoproteomics datasets were performed using Perseus (*version* 1.5.5.3)[70]. During data processing, reverse and contaminant database hits and candidates identified only by site were filtered out. For differential quantitative proteomics analyses, categorical annotation (NS/SD/RS) was applied to group reporter ion intensities, values were $\log_2$ transformed, and were normalized by "subtract mean (column-wise)" in each TMT reporter ion channel. Proteins groups were filtered for valid values (at least 80% in each group). Unpaired *t*-tests followed by false discovery rate (FDR) analysis was performed to compare sleep treatments groups.

**Gene ontology (GO) analysis**. Significantly and differentially expressed transcripts and proteins across sleep treatment groups were subjected to GO analysis. We performed an overrepresentation analysis[71] that was implemented in clusterProfiler[72]. Significantly enriched GO terms were visualized as dot plots and enrichment maps.

**Network analysis**. To investigate the association between significant and differentially expressed global proteins in sleep treatment groups from astrocytes and neurons we extracted protein associations from the Search Tool for the Retrieval of Interacting Genes or Proteins (STRING) database[49]. STRING identified 462/463 significant proteins and returned 863 interactions in astrocytes (protein-protein interaction (PPI) enrichment *P*-value < 0.001), 695/701 significant proteins and 1702 interactions in neurons (PPI enrichment *P*-value < 0.001) at a confidence cut-off value 0.9 (the highest confidence level in STRING).

**Analysis of RNAscope in situ hybridization images**. RNA markers were quantified based on dots quantified within cortical and hypothalamic areas. We counted the cells from three brains per sleep treatment group[68]. We counted the almost entire cortex (1500 × 800 μm, 1200 × 800 μm and 900 × 800 μm) and hypothalamus by defined region of interests (ROIs) (Fig. 6a). The number of transcripts per cell was determined by using Visiopharm's image analysis algorithm.

**Plasma corticosterone analysis**. Blood samples were collected in tubes on ice containing EDTA and later centrifuged at 4 °C. Plasma concentrations of corticosterone were measured employing radioimmunoassay kits (Millipore, Billerica, USA).

**Statistics and reproducibility**. All experimental subjects are biological replicates. For single-cell sequencing, dissections of the same anatomical region were pooled from three animals for the preparation of single-cell suspensions for each group. GraphPad Prism 8 or R software was used to perform statistical tests. Kruskal–Wallis tests were performed to compare sleep treatments groups. Following two-way analysis of variance (ANOVA), Fisher's LSD tests were performed for comparisons. Repeated measures tests were performed for same-subject comparisons. *P* < 0.05 was considered as significant. For transcriptomics and proteomics analyses, we used FDR < 0.1, unless indicated otherwise. Detailed sample sizes, statistical tests, and results are reported in the figure legends and Supplementary Data 1.

**Reporting summary**. Further information on research design is available in the Nature Research Reporting Summary linked to this article.

## Data availability
The single cell RNA-Seq datasets generated for this study have been deposited in the in the Gene Expression Omnibus (accession number: GSE137665). This dataset is accessible via: https://www.ncbi.nlm.nih.gov/geo/query/acc.cgi?acc=GSE137665 (please enter token to access: urepqsyorravfml). The mass spectrometry proteomics and phosphoproteomics data described in this article are deposited to the ProteomeXchange Consortium via the PRIDE partner repository with the dataset identifier PXD018334. Reviewer account details are: Username: reviewer55333@ebi.ac.uk, Password: mAKgsWFP. The source data values underlying Figs. 1b, 3b–e, 4b–e, 5b–e, 6b–f, and 8b, c can be found in Supplementary Data 2.

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

## Acknowledgements

A.B.R. acknowledges funding from the Perelman School of Medicine, University of Pennsylvania, the Institute for Translational Medicine and Therapeutics (ITMAT) at the University of Pennsylvania. This work was supported also by NIH DP1DK126167 and R01GM139211 (A.B.R).

## Author contributions

A.B.R., P.K.J. and U.K.V. conceived and designed the experiments. P.K.J. and M.N. performed sleep phenotyping by EEG/EMG analysis. P.K.J. performed mouse sleep deprivation experiments and tissue dissections. P.K.J. and U.K.V. performed single-cell isolation and library preparation. U.K.V., P.K.J. and A.B.R. analyzed single-cell tran-scriptomics data. P.K.J and U.K.V performed RNAScope in situ hybridization analysis. P.K.J. and U.K.V. performed astrocyte and neuron isolation from the mice cortex. S.R. and U.K.V. performed quantitative proteomics and phosphoproteomics. S.R., P.K.J., U.K.V. and A.B.R. analyzed proteomics data. A.B.R. supervised the entire study and secured funding. The manuscript was written by P.K.J. and A.B.R. with input from U.K.V. All authors agreed on the interpretation of data and approved the final version of the manuscript.

## Competing interests

The authors declare no competing interests.
