## [Peer Review File · Communications Biology]

Reviewers' comments:

Reviewer #1 (Remarks to the Author):

In this study, Jha and colleagues investigate the molecular signatures of sleep using cell specific techniques. As recent research has revealed, sleep or its loss has differential impacts on cells depending upon both brain region and cell type. Given the prevalence of sleep deprivation in teenagers and adults, this study is very timely as it is important to identify the molecular ramifications of sleep and sleep loss. The authors interrogated the molecular signatures of sleep in a cell type specific manner using single cell RNA sequencing and phosphoproteomics in three brain regions, the brainstem, cortex and hypothalamus. The results of this study presented in this well-written and clear manuscript emphasize the growing conclusion that sleep deprivation impacts has transcriptional and translational impacts dependent upon cell type. However, the manuscript appears to ignore recent research in this area with a lack of key relevant publications from 2020 and 2021 and two notable Science papers from 2019, particularly those that examine cell type specific changes with sleep deprivation. This gap in references to recent studies presents a skewed picture to the reader and demonstrates a willful lack of scholarship by the authors. The authors look at areas such as the brainstem which are understudied for analysis provides interesting information. While the information generated from this study will contribute to our understanding of the impacts of sleep deprivation and sleep on gene regulation, there are some concerns regarding methodology and experimental design which may lessen the impact of the results.

Comments:

1. Methods – Single Cell RNA seq. It is unclear as to whether the number of cells used in the study (~29,000) are sufficient to make conclusion for 3 brain regions and 3 different conditions – this seems to be underpowered representing approximately 3200 cells per group.
2. Methods: The authors should also indicate what the approximate percent of neurons to astrocytes to other cell types was in the Results as this reflects the technical details of the procedure, particularly as the authors comment that astrocytes display larger alterations in tan neurons in the brainstem and the hypothalamus. Did the authors sequence the expected proportion of neurons from these areas?
3. Methods and Results” Why was there such a difference in quantifying proteins from astrocytes and neurons? Was there a technical reason underlying this?
4. Methods and Results – The RNAScope studies appear to be done for a very small number of genes, so they do not really provide a validation of the RNaseq results (3 genes in the cortex and 2 genes in the hypothalamus). Why was no RNAScope done for the brainstem.
5. Introduction, Results and Discussion – there appears to be a gap in the presentation of the literature, particularly for sleep deprivation and gene regulation studies done recently (2019-2021) including transcriptomics studies, cell-type specific studies and proteomics/phosphoproteomics studies.

Consequently, the authors make statements and claims that are untrue such as on lines 236-238 “Moreover, in the context of sleep, no studies have been performed to investigate sleep need regulation within specific cell types in the cortex (or any other brain region).” And a further statement which is untrue in the Discussion lines 314-316 which states “This had not been appreciated before because bulk methods by their nature average out the individual contributions of diverse cell types in each brain region.”

Ignoring the research on sleep/wake and 6 h sleep deprivation from the Science papers published by the labs of Steve Brown and Maria Robles (PMID: 31601740 and PMID: 31601739) hurts the Discussion and presents a very skewed picture to readers as these labs clearly published that the changes in the proteome were driven by sleep need, with sleep-wake cycles changing synaptic

phosphorylation.

Minor Comments:

1. Methods – the authors do not provide the rationale for the 12 hour duration of sleep deprivation. Many prior studies have used 5 – 6 hours of sleep deprivation as this has been shown sufficient to inhibit long-term memory and induce cellular and structural changes in neurons. The rationale for choosing the longer period of sleep deprivation needs to be made clear.
2. Methods – similarly, the rationale for examining the brain regions chosen for analysis should be provided
3. No rational is provided for the genes the author chose to validate using RNAScope.
4. Figure 1 – the error bars do not appear clear on the table B – data points for individual animals should be shown

Reviewer #2 (Remarks to the Author):

This work seeks to investigate the molecular underpinnings of sleep homeostasis using scRNA-seq analysis and proteomics in the hypothalamus, cortex, and brainstem. Comparing transcriptomics, proteomics, phosphoproteomics, and gene regulatory networks (GRNs), the group concluded that sleep need regulates transcriptional, translational, and post-translational responses in a cell specific manner. Sleep deprivation regulates astrocyte-neuron cross talk and enhances specific sets of transcription factors (TFs) in a region-specific manner, as well as cell type specific phosphorylation. This study is overall well-executed, contains a considerable amount of data, and will provide a useful resource to researchers studying the molecular mechanisms controlling sleep. Suggestions for improvement are listed below:

Specific Comments:

1. The 24 hr RS is extremely long, and more than sufficient to dissipate all the sleep pressure accumulated during the 12hr SD. Including data additional RS timepoints (particularly in the 4-8hr range) would be extremely informative, as this would allow identification of molecular changes directly responsive to sleep pressure changes.
2. The number of technical and biological replicates performed for scRNA-Seq and proteomic analysis needs to be stated.
3. Is the automated sweeping bar used here for SD a greater psychological stressor than alternative approaches (e.g. gentle handling)? Can previous studies be cited to support this?
4. Graphs are often difficult to read, and are not clearly labeled, and/or captions are difficult to understand. Font size often very small, and too many individual items are labeled. Examples include Figure 1C,D (poorly labeled); Figures 2C-E, 3C-E (small font size and too many items labeled);
5. In every case where neuronal subtype clusters are shown, these should be identified based on molecular markers, and an effort made to match them to previously identified cell types from that brain region.
6. Quantification of expression changes measured by in situ hybridization should be included.
7. Clearer explanations of how candidate genes were selected for in situ hybridization need to be provided.

Additional questions:

1. In line 101: "Animals had 93% sleep loss in the first six hours of sleep restriction (ZT0-6) and 88% in the later six hours (ZT6-12) (Fig. 1B)." How was sleep loss calculated here? Total time in NREM+REM? Sleep bout number? Delta power vs. baseline?
2. In line 178 "These results show that alteration of sleep need causes profound and functionally distinct gene expression changes within individual cells residing in different brain regions, which was not known before." Many studies have shown cell type-specific gene expression changes in response to changes in sleep pressure (e.g. Liu, et al. 2017; Puentes-Mestril, et al. 2021, etc). These studies should be cited, and this sweeping claim scaled back.
3. In line 220 "Importantly, out of all the TFs regulating gene expression in cells from sleep-deprived animals, we did not find any TF that was common to any two brain regions." This likely simply reflects the fact that the number of cells (~10,000/region over 3 conditions) and experimental replicates (n=1?), and that the sample is relatively underpowered. Though this is not a criticism of the study itself, which is an important exploratory first step, interpretation of these results should be tempered.
4. In line 237, "Moreover, in the context of sleep, no studies have been performed to investigate sleep need regulation within specific cell types in the cortex (or any other brain region)". Again, this is not true as stated here. Are the investigators specifically referring to scRNA-Seq or proteomic studies? If so, then please be specific.
5. In line 359, "We found that 22% of neuronal proteins responded to sleep perturbation compared to only 3% in astrocytes". This is a very interesting point, and would benefit from further elaboration.

Point by point responses to reviewer comments

Single-cell transcriptomics and cell-specific proteomics reveals molecular signatures of sleep

Response to Reviewer 1

In this study, Jha and colleagues investigate the molecular signatures of sleep using cell-specific techniques. As recent research has revealed, sleep or its loss has differential impacts on cells depending upon both brain region and cell type. Given the prevalence of sleep deprivation in teenagers and adults, this study is very timely as it is important to identify the molecular ramifications of sleep and sleep loss. The authors interrogated the molecular signatures of sleep in a cell type-specific manner using single-cell RNA sequencing and phosphoproteomics in three brain regions, the brainstem, cortex, and hypothalamus. The results of this study presented in this well-written and clear manuscript emphasize the growing conclusion that sleep deprivation impacts have transcriptional and translational impacts dependent upon cell type. However, the manuscript appears to ignore recent research in this area with a lack of key relevant publications from 2020 and 2021 and two notables.

Science papers from 2019, particularly those that examine cell-type-specific changes with sleep deprivation. This gap in references to recent studies presents a skewed picture to the reader and demonstrates a wilful lack of scholarship by the authors. The authors look at areas such as the brainstem which are understudied for analysis provides interesting information. While the information generated from this study will contribute to our understanding of the impacts of sleep deprivation and sleep on gene regulation, there are some concerns regarding methodology and experimental design which may lessen the impact of the results.

We thank the reviewer for an in-depth evaluation of our manuscript and for their insightful remarks and suggestions. In response to these suggestions, we have incorporated the results of some new analyses and analytical discussion in the revised manuscript to significantly improve the manuscript. Responses to individual comments are below.

Comment 1. Methods – Single Cell RNA seq. It is unclear as to whether the number of cells used in the study (~29,000) is sufficient to make a conclusion for 3 brain regions and 3 different conditions – this seems to be underpowered representing approximately 3200 cells per group.

We thank the reviewer for seeking clarification on the number of cells sequenced per brain area per sleep condition, and whether these numbers are sufficient to represent the respective brain areas. We performed a literature search to determine metrics for scRNA-seq studies of brain areas from young adult mouse brains (i.e. the same tissue we assayed). We noted that there is significant variation in the metrics used for each of these studies, and no clear consensus exists on the appropriate number of cells to use in this circumstance.

For example, Chen et al 2017 (Chen et al., 2017) used a total of 14,000 cells to generate the clusters for the study of the transcriptional responses in the hypothalamus, comparing two groups. They narrowed it down to 3,319 cells for the generation of final clustering.

On the other hand, Mickelsen and colleagues (Mickelsen et al., 2019) used 3,784 cells from the male mouse brain and 3,434 cells from female mouse brain for initial clustering. Some of these studies are summarized below (Table 1). Other studies used fewer cells.

Together, we sequenced a total of 29,051 cells, and we used ~ 3,300 cells per group. These metrics make our study comparable with other published studies of adult mouse brain that

Publication	Brain region	Cells sequenced and QC
Rossi et al 2019, PMID: 31249056	Lateral hypothalamus	Total cell sequenced: 20,194, two groups, 20,194 cells for cluster
Mickelsen et al 2019, PMID: 30858605	Lateral hypothalamus	Total cell sequenced: 7218 (3,784 males and 3,434 females)
Chen et al 2017, PMID: 28355573	Hypothalamus	Total cell sequenced: 14000, two groups, 3319 cells for cluster
Tasic et al 2016, PMID: 26727548	Cerebral cortex	Total cell sequenced: 1739, 1679 cells for cluster

we are aware of (Table 1).

Table 1: Comparison of the number of cells sequenced in published single-cell sequencing of adult mouse brain

Comment 2. Methods: The authors should also indicate what the approximate percent of neurons to astrocytes to other cell types was in the Results as this reflects the technical details of the procedure, particularly as the authors comment that astrocytes display larger alterations than neurons in the brainstem and the hypothalamus. Did the authors sequence the expected proportion of neurons from these areas?

Based on the reviewer's suggestions, we prepared a table showing the percentages of different cell types sequenced and analyzed in all three brain areas (Table 2). We have included these results as Table S11, and mention the proportion of astrocytes and neurons in the brainstem and hypothalamus in the Results section of the revised manuscript.

Brain area	Cell Type	NS	SD	RS
Brainstem	Endothelial	28.60	25.19	22.88
	Microglia	27.05	20.38	11.94
	VSMCA	11.29	12.05	12.34
	Astrocyte	9.29	14.63	15.09
	Pericyte	8.73	8.83	6.04
	EPC	4.81	8.15	17.73
	Neuron	3.69	4.28	3.57
	Macrophage	3.52	3.75	4.47
	Oligo	1.94	1.23	1.49
	HbVC	1.07	1.50	2.25
	Unknown	0.03	0.00	2.19
Cortex	Astrocyte	40.60	39.49	39.49
	Neuron	25.33	29.15	29.15
	Microglia	17.83	8.10	8.10
	Endothelial	7.42	14.59	14.59
	VSMCA	3.85	4.29	4.29
	Macrophage	2.10	1.04	1.04
	HbVC	1.13	1.51	1.51
	Oligo	0.97	1.13	1.13
	Pericyte	0.78	0.69	0.69
Hypothalamus	Neuron	64.47	50.78	77.90
	Astrocyte	16.04	18.85	12.50
	Endothelial	6.47	7.69	2.31
	Microglia	4.92	6.10	1.26
	EPC	3.51	8.61	4.54
	VSMCA	2.36	3.10	0.28
	Pericyte	1.72	3.73	0.95
	HbVC	0.52	1.15	0.26

Table 2: Percentage proportions of different cell types in three sleep conditions (NS=normal sleep; SD=sleep deprived; RS=recovery sleep) in brainstem, cortex, and hypothalamus

A higher proportion of cells were classified as astrocytes than neurons in both the brainstem and cortex, and we observed larger shifts in gene expression in brainstem astrocytes and cortical neurons. On the other hand, we identified ~65% of cells as neurons and only ~15 % as astrocytes in the hypothalamus. The alteration of gene expression does not reflect the proportions of the numbers of cells. Our results are consistent with previous studies of having a higher proportion of neurons in the hypothalamus, and non-neuronal cells in the cortex (Chen et al., 2017; Hrvatin et al., 2018; Moffitt et al., 2018). We could not make comparisons for the brainstem since comparable single-cell studies in adult mouse brain are not available.

Comment 3. Methods and Results” Why was there such a difference in quantifying proteins from astrocytes and neurons? Was there a technical reason underlying this?

We thank the reviewer for their query. We identified 2,454 proteins in astrocytes and 1,401 in neurons. We isolated neurons and astrocytes from the cortex by magnetic affinity cell sorting (please see Materials and Methods). Given that cortical neurons are highly branched and interconnected, and that only neuronal soma are enriched for in this affinity purification method, these isolations lead to the loss of axons and distal dendrites (which contain proteins), and this could be reflected in the lower abundance of proteins in neurons. By contrast, all of the astrocyte's cytoplasm is captured, leading to a higher number of proteins identified (and a higher abundance of protein in general).

Comment 4. Methods and Results – The RNAScope studies appear to be done for a very small number of genes, so they do not really provide a validation of the RNASeq results (3 genes in the cortex and 2 genes in the hypothalamus). Why was no RNAScope done for the brainstem.

In our sc-RNAseq study, we aimed to understand how the change in sleep-wake states affects gene expression patterns within the cells of the brainstem, cortex, and hypothalamus. Therefore, we did not select particular brain regions to validate single-cell gene expression by RNAScope. Rather, we looked at transcripts that were potentially interesting (and that we may wish to follow-up in future studies), and which changed in different brain regions during sleep deprivation. We did not validate genes in the brainstem as it would be difficult to select sub-anatomical zones to validate the RNAseq results given the anatomical complexity of brainstem nuclei and the overall length of this brain area – it would require significant amount of sectioning to home in on the relevant region expressing the particular transcript.

Furthermore, we do not believe that every brain region needs to be validated, as we assessed two and found no differences in between RNAseq and RNAScope in those regions. There would be no reason to think that the same would not be true for brainstem. Moreover, we believe a total of five RNAScope validations is in line with previous literature [e.g (Ximerakis et al., 2019), especially considering that we assayed three sleep conditions for each gene/brain region. Most RNAScope validations simply look for whether a transcript is expressed in a section (i.e. one condition) rather than in multiple conditions (three in our case), resulting in more assessments in our study in total.

Comment 5. Introduction, Results and Discussion – there appears to be a gap in the presentation of the literature, particularly for sleep deprivation and gene regulation studies done recently (2019-2021) including transcriptomics studies, cell-type-specific studies and proteomics/phosphoproteomics studies.

Consequently, the authors make statements and claims that are untrue such as on lines 236-238 “Moreover, in the context of sleep, no studies have been performed to investigate sleep need regulation within specific cell types in the cortex (or any other brain region).” And a further statement which is untrue in the Discussion lines 314-316 which states “This had not been appreciated before because bulk methods by their nature average out the individual contributions of diverse cell types in each brain region.”

Ignoring the research on sleep/wake and 6 h sleep deprivation from the Science papers published by the labs of Steve Brown and Maria Robles (PMID: 31601740 and PMID: 31601739) hurts the Discussion and presents a very skewed picture to readers as these labs clearly published that the changes in the proteome were driven by sleep need, with sleep-wake cycles changing synaptic phosphorylation.

We thank the reviewer for these comments. We selected some recent studies based on their similarity to our experimental paradigm and compared them with our study. We attempted to assign the cell-specificity to previous bulk RNAseq data. Most recent sleep transcriptomics studies were performed on the cortex (Diessler et al., 2018; Gerstner et al., 2016; Scarpa et al., 2018). In general, we did not find a good overlap of differentially expressed genes (DEGs) among these studies themselves, and therefore grouped all DEGs detected from any of these studies into a single group (“Cumulative bulk RNAseq”) to compare with our scRNA-seq data.

We found some overlap in astrocytes, neurons, endothelial cells, and microglia (Fig. S7A of the manuscript). The majority of other DEGs were not detected in the scRNA-seq dataset (Table S4 in manuscript). We also compared DEGs in the hypothalamus (Scarpa et al., 2018) with our data (Fig. S7B of the manuscript, Table S4 in manuscript). Again, this analysis assigned some of these previously-reported sleep-driven alterations in hypothalamic astrocytes, neurons, endothelial cells, and microglia.

Furthermore, we compared our cell-specific proteomics dataset with published whole-brain proteomics (Wang et al., 2018) data. Wang et al. 2018 concluded that the whole brain proteome remains globally stable, but phosphoproteomes showed significant changes in the response to sleep treatments. In contrast, our cell-specific global proteomics data in cortex reveals significant alteration in protein abundance in astrocytes (93/2,454, 3.7%; 1.5-fold cutoff, FDR < 0.1) and neurons (318/1,401, 22.7%; 1.5-fold cut-off, FDR < 0.1) following sleep treatments. This disparity is likely explained by reciprocal changes in different brain areas and/or cell types that will be “masked” by bulk cell proteomics, particularly in whole brain samples, and highlights the utility of cell- and region-specific analyses such as those performed in our study.

For phosphoproteomics, we compared differentially abundant (FDR < 0.1) phosphorylated proteins from the whole brain (Wang et al., 2018) with our astrocyte and neuron datasets. This analysis enabled us to assign phosphorylation of proteins reported by Wang et al. 2018 in whole brain samples to neurons and astrocytes (Fig. S7C, and Table S10 in manuscript), which could not be determined purely from their dataset.

The molecular alterations reported after sleep deprivation could arise due to extended wake, elevated sleep pressure, or the cumulative effect of both. The Science papers from the Brown and Robles labs elegantly describe how elevated sleep pressure over the day-night cycle affects daily rhythms of forebrain synaptoneurosomal transcription, translation, and post-translational changes. Synaptoneurosomes are resealed vesicles or isolated terminals that break away from axon terminals when cortex is homogenized. They retain pre- and postsynaptic characteristics, which makes them useful in the study of synaptic transmission. Thus, these preparations are not representative of the entire neuron (and in particular exclude the soma and the majority of the axons – i.e. the majority of the cell content that we assayed in our study). Thus, we cannot compare our transcriptomics

and proteomics results with these studies because there will be (by design) a bias towards synapse transcripts and proteins in their datasets, whereas we did not select out any particular compartment of neurons. Nevertheless, we have discussed these studies in the second last paragraph of the Discussion of the revised submission to acknowledge and include relevant literature. As suggested, we also reworded lines 236-238 and 314-316 of the manuscript.

Minor Comments:

Comment 1. Methods – the authors do not provide the rationale for the 12-hour duration of sleep deprivation. Many prior studies have used 5 – 6 hours of sleep deprivation as this has been shown sufficient to inhibit long-term memory and induce cellular and structural changes in neurons. The rationale for choosing the longer period of sleep deprivation needs to be made clear.

We apologize for not being clear with our 12 hours of sleep deprivation model in the first submission. In our study, we aimed to study the molecular responses of sleep need at the transcriptional, translational, and post-translational levels. We hypothesized that compared to the 5-6 hours of sleep deprivation timeline, 12 hours of sleep deprivation should provide sufficient time for transcriptional changes to become apparent at the protein level. Further, we designed our study to collect the brain samples of the three sleep treatment groups at the same time of day to avoid the confound of time-of-day variation in gene/proteins abundance within the cells (see Fig. 1A). We have explained better the rationale for 12 hours of sleep deprivation in the Sleep deprivation sub-section of the Materials and Methods section in the revised manuscript.

Comment 2. Methods – similarly, the rationale for examining the brain regions chosen for analysis should be provided

The main brain areas that regulate sleep-wake states includes the brainstem, cerebral cortex, thalamus, and hypothalamus (Rosenwasser, 2009; Saper and Fuller, 2017). In the present study, we looked for the molecular responses of sleep need at the individual cell level in the brainstem, cortex, and hypothalamus of the mouse brain. We mentioned this in the second paragraph of the Introduction and Sleep phenotyping and experimental design section of the Results. To clarify this further, we have included the rationale of the selection of the brain areas for our study in the Sleep deprivation sub-section of the Materials and Methods section in the revised manuscript as suggested.

Comment 3. No rationale is provided for the genes the author chose to validate using RNAScope.

*We thank the reviewer for seeking the rationale for our selection of candidate genes for the validation. We first sorted the genes based on FDR and fold change, and then based on the availability of RNAScope probes we opted for *Mt1*, *Tsc22d2*, and *Gjg6* in the cortex and *Mt1* and *Atp1b2* in the hypothalamus. We also chose these genes because we may follow-up on some of these in our future work. We have included these details in the RNAscope in situ hybridization sub-section of the Materials and Methods section in the revised manuscript.*

Comment 4. Figure 1 – the error bars do not appear clear on the table B – data points for individual animals should be shown

We apologize for this omission. We have revised Fig. 1B to show data points for individual animals, and error bars, as suggested.

Response to Reviewer 2

This work seeks to investigate the molecular underpinnings of sleep homeostasis using scRNA-seq analysis and proteomics in the hypothalamus, cortex, and brainstem. Comparing transcriptomics, proteomics, phosphoproteomics, and gene regulatory networks (GRNs), the group concluded that sleep need regulates transcriptional, translational, and post-translational responses in a cell specific manner. Sleep deprivation regulates astrocyte-neuron cross talk and enhances specific sets of transcription factors (TFs) in a region-specific manner, as well as cell type specific phosphorylation.

This study is overall well-executed, contains a considerable amount of data, and will provide a useful resource to researchers studying the molecular mechanisms controlling sleep. Suggestions for improvement are listed below:

We appreciate the reviewer's comprehensive analysis, encouraging remarks, and valuable suggestions to improve the manuscript. In this revised submission, we performed additional analyses, quoted suggested references, and modified the manuscript accordingly. We address the specific queries and comments below.

Specific Comments:

Comment 1. The 24 hr RS is extremely long, and more than sufficient to dissipate all the sleep pressure accumulated during the 12hr SD. Including data additional RS timepoints (particularly in the 4-8hr range) would be extremely informative, as this would allow identification of molecular changes directly responsive to sleep pressure changes.

We agree with the reviewer's point that 24 hours of recovery is long, and it could lead to dissipation of sleep pressure accumulated during the 12 hours of sleep deprivation. Considering the available resources for scRNA-seq, we collected the tissues at a single time of day (ZT12). We performed sleep deprivation between ZT0 to ZT12 and collected the tissues at ZT12 for NS, SD, and RS groups to limit the confounds due to time-of-day variation. This represented the most efficient use of resources. If we had assayed after 4-8 hours of recovery sleep there would be a significant time-of-day difference between sleep-deprived and recovery groups, which would confound sleep-dependent change with time-of-day variation (due to the circadian cycle). In an ideal scenario, it would be useful to assay multiple recovery time points (e.g 4h, 6h or 8h) along with parallel controls. However, this would require significantly more resources – approx. 3x the number of scRNAseq samples.

Comment 2. The number of technical and biological replicates performed for scRNA-Seq and proteomic analysis needs to be stated.

We thank the reviewer for seeking clarity over the technical and biological replicates for scRNA-Seq and proteomic analysis. For scRNA-seq experiments, we pooled samples from

n=3 animals for each sleep treatment group to prepare single-cell suspensions. For example, we pooled the cortex from n=3 animals in the groups: n=3 normal sleep (NS_Cortex), n=3 sleep-deprived (SD_Cortex), and n=3 recovery sleep (RS_Cortex). Likewise, for the hypothalamus and brainstem, we did the same. Therefore, in total, we performed scRNA-Seq on 9 samples, with a single technical replicate for each. For proteomics analysis, we dissected the cortex from n=3 animals for each sleep treatment group. Therefore, we had three biological replicates per group. We have included this information in the Materials and Methods section of the revised manuscript.

Comment 3. Is the automated sweeping bar used here for SD a greater psychological stressor than alternative approaches (e.g. gentle handling)? Can previous studies be cited to support this?

We thank the reviewer for this comment. It is accepted in the field that most sleep deprivation methods act as a stressor for experimental animals. We opted automated sweeping bar method for sleep deprivation to minimize the stress levels of animals as manual handling or forced locomotion are known to increase plasma corticosterone levels in rodents (Meerlo et al., 2002; Mongrain et al., 2010). We also validated this by performing the plasma corticosterone assay of our sleep treatment group. We did not see higher levels of plasma corticosterone in the animals after sleep deprivation by the sweeping bar method (Fig. 1). We have included this in the revised submission as Fig. S1E and updated the text accordingly.

Fig. 1. Plasma corticosterone level (n = 6). Data are presented as mean \pm SEM.

Comment 4. Graphs are often difficult to read, and are not clearly labeled, and/or captions are difficult to understand. Font size often very small, and too many individual items are labeled. Examples include Figure 1C,D (poorly labeled); Figures 2C-E, 3C-E (small font size and too many items labeled);

We thank the reviewer for highlighting this. We have revised the graphs and figures as suggested. Fig. 1C, D have been re-labelled, and Figs. 3C-E, 4C-E, and 5C-E were re-plotted. We also manually annotated gene names in those figures for clearer visualization by readers.

Comment 5. In every case where neuronal subtype clusters are shown, these should be identified based on molecular markers, and an effort made to match them to previously identified cell types from that brain region.

We based classifications of cell type on previously published literature. However, when we examined more closely the markers found by published scRNA-seq studies of the adult mouse brain (Chen et al., 2017; Hrvatin et al., 2018; Mickelsen et al., 2019; Rossi et al., 2019; Saunders et al., 2018; Tasic et al., 2016; Zeisel et al., 2015), we found poor overlap in markers for each cell type. There was little or no consistency in defining cell types, especially neuronal subtypes. Therefore, in our initial submission, we could not further subtype the neuronal cells. As suggested by reviewer, we attempted to classify neurons in all three brain areas using as many markers as possible. In the brainstem and cortex, we classified neuronal clusters into “Glutamatergic neurons” and “Other neurons” (lack of specific marker). Similarly, in the hypothalamus, we classified neuronal clusters into

GABAergic, Other, and other Inhibitory neurons. We have updated the UMAP plots (Figs. 3A, 4A, and 5A) and modified the text accordingly in the revised version of the manuscript.

Comment 5. Quantification of expression changes measured by in situ hybridization should be included.

We have revised the RNAscope in situ hybridization plots by replacing Z-score mRNA counts per cell with expression changes as mRNA counts per cell.

Comment 5. Clearer explanations of how candidate genes were selected for in situ hybridization need to be provided.

We thank the reviewer for seeking the rationale for our selection of candidate genes for the validation. We first sorted the genes based on FDR and fold changes then based availability of RNAscope probes we opted for *Mt1*, *Tsc22d2*, and *Gjg6* in the cortex and *Mt1* and *Atp1b2* in the hypothalamus. We also chose these genes because we may follow-up on some of these in our future work. We have included these details in the RNAscope in situ hybridization sub-section of the Materials and Methods section in the revised manuscript.

Additional questions:

1. In line 101: “Animals had 93% sleep loss in the first six hours of sleep restriction (ZT0-6) and 88% in the later six hours (ZT6-12) (Fig. 1B).” How was sleep loss calculated here? Total time in NREM+REM? Sleep bout number? Delta power vs. baseline?

We apologize for not being clear about the percentage of sleep restriction. We measured the effectiveness of our sleep deprivation method in terms of the percentage of wakefulness during 12 hours of sleep restrictions. We have updated the figure with n=6 animals after further analysis. We now report an average of 91.5% wakefulness during the first six hours of sleep restriction (ZT0-6) and 86.6% wakefulness during the remaining six hours (ZT6-12) (Fig. 1B of the manuscript). We hope this addresses the reviewer’s query adequately.

Comment 2. In line 178 “These results show that alteration of sleep need causes profound and functionally distinct gene expression changes within individual cells residing in different brain regions, which was not known before.” Many studies have shown cell type-specific gene expression changes in response to changes in sleep pressure (e.g. Liu, et al. 2017; Puentes-Mestral, et al. 2021, etc). These studies should be cited, and this sweeping claim scaled back.

We apologize for missing these studies in our literature search, and we are grateful for the reviewer bringing these to our attention. We have included these studies in our references and modified the above line in the revised submission.

Comment 3. In line 220 “Importantly, out of all the TFs regulating gene expression in cells from sleep-deprived animals, we did not find any TF that was common to any two brain regions.” This likely simply reflects the fact that the number of cells (~10,000/region over 3 conditions) and experimental replicates (n=1?), and that the sample is relatively underpowered. Though this is not a criticism of the study itself, which is an important exploratory first step, interpretation of these results should be tempered.

We thank the reviewer for these comments, and have removed this statement to temper our interpretation of the results accordingly.

Comment 4. In line 237, “Moreover, in the context of sleep, no studies have been performed to investigate sleep need regulation within specific cell types in the cortex (or any other brain region)”. Again, this is not true as stated here. Are the investigators specifically referring to scRNA-Seq or proteomic studies? If so, then please be specific.

We apologize for not being clear with this sentence. We refer to our cell-specific proteomics (neuron and astrocyte) studies in the context of sleep that in our knowledge it is the first of its kind. We specified this line in the revised version of the manuscript.

Comment 5. In line 359, “We found that 22% of neuronal proteins responded to sleep perturbation compared to only 3% in astrocytes”. This is a very interesting point, and would benefit from further elaboration.

We reported larger shifts in proteins expressed in neurons than in astrocytes despite a lower abundance of proteins in neurons. Keeping a relatively stringent cut-off (1.5-fold cut-off, FDR < 0.1), we reported 22% of neuronal proteins and only 3% astrocytes proteins responded to sleep treatments. This indicates astrocytes are less responsive to acute sleep pressure than neurons. We hypothesized that under physiological challenges such as sleep loss/prolonged wakefulness relatively stable astrocytes could extend their support to neurons to maintain synaptic and metabolic homeostasis. We have discussed this in the 6th paragraph of the Discussion in the manuscript and added a sentence after.

References

Chen, R., Wu, X., Jiang, L., and Zhang, Y. (2017). Single-Cell RNA-Seq Reveals Hypothalamic Cell Diversity. *Cell Reports* 18, 3227–3241.

Diessler, S., Jan, M., Emmenegger, Y., Guex, N., Middleton, B., Skene, D.J., Ibberson, M., Burdet, F., Götz, L., Pagni, M., et al. (2018). A systems genetics resource and analysis of sleep regulation in the mouse. *Plos Biol* 16, e2005750.

Gerstner, J.R., Koberstein, J.N., Watson, A.J., Zaperro, N., Risso, D., Speed, T.P., Frank, M.G., and Peixoto, L. (2016). Removal of unwanted variation reveals novel patterns of gene expression linked to sleep homeostasis in murine cortex. *Bmc Genomics* 17, 727.

Hrvatín, S., Hochbaum, D.R., Nagy, M.A., Cicconet, M., Robertson, K., Cheadle, L., Zilionis, R., Ratner, A., Borges-Monroy, R., Klein, A.M., et al. (2018). Single-cell analysis of experience-dependent transcriptomic states in the mouse visual cortex. *Nat Neurosci* 21, 120–129.

Meerlo, P., Koehl, M., Borghot, K.V.D., and Turek, F.W. (2002). Sleep Restriction Alters the Hypothalamic-Pituitary-Adrenal Response to Stress: Sleep restriction and HPA axis reactivity. *J Neuroendocrinol* 14, 397–402.

Mickelsen, L.E., Bolisetty, M., Chimileski, B.R., Fujita, A., Beltrami, E.J., Costanzo, J.T., Naparstek, J.R., Robson, P., and Jackson, A.C. (2019). Single-cell transcriptomic analysis of the lateral

hypothalamic area reveals molecularly distinct populations of inhibitory and excitatory neurons. *Nat Neurosci* 22, 642–656.

Moffitt, J.R., Bambah-Mukku, D., Eichhorn, S.W., Vaughn, E., Shekhar, K., Perez, J.D., Rubinstein, N.D., Hao, J., Regev, A., Dulac, C., et al. (2018). Molecular, spatial, and functional single-cell profiling of the hypothalamic preoptic region. *Sci New York N Y* 362.

Mongrain, V., Hernandez, S.A., Pradervand, S., Dorsaz, S., Curie, T., Hagiwara, G., Gip, P., Heller, H.C., and Franken, P. (2010). Separating the contribution of glucocorticoids and wakefulness to the molecular and electrophysiological correlates of sleep homeostasis. *Sleep* 33, 1147–1157.

Rosenwasser, A.M. (2009). Functional neuroanatomy of sleep and circadian rhythms. *Brain Res Rev* 61, 281–306.

Rossi, M.A., Basiri, M.L., McHenry, J.A., Kosyk, O., Otis, J.M., Munkhof, H.E. van den, Bryois, J., Hübel, C., Breen, G., Guo, W., et al. (2019). Obesity remodels activity and transcriptional state of a lateral hypothalamic brake on feeding. *Science* 364, 1271–1274.

Saper, C.B., and Fuller, P.M. (2017). Wake–sleep circuitry: an overview. *Curr Opin Neurobiol* 44, 186–192.

Saunders, A., Macosko, E.Z., Wysoker, A., Goldman, M., Krienen, F.M., Rivera, H. de, Bien, E., Baum, M., Bortolin, L., Wang, S., et al. (2018). Molecular Diversity and Specializations among the Cells of the Adult Mouse Brain. *Cell* 174, 1015-1030.e16.

Scarpa, J.R., Jiang, P., Gao, V.D., Fitzpatrick, K., Millstein, J., Olker, C., Gotter, A., Winrow, C.J., Renger, J.J., Kasarskis, A., et al. (2018). Cross-species systems analysis identifies gene networks differentially altered by sleep loss and depression. *Sci Adv* 4, eaat1294.

Tasic, B., Menon, V., Nguyen, T.N., Kim, T.K., Jarsky, T., Yao, Z., Levi, B., Gray, L.T., Sorensen, S.A., Dolbeare, T., et al. (2016). Adult mouse cortical cell taxonomy revealed by single cell transcriptomics. *Nat Neurosci* 19, 335–346.

Wang, Z., Ma, J., Miyoshi, C., Li, Y., Sato, M., Ogawa, Y., Lou, T., Ma, C., Gao, X., Lee, C., et al. (2018). Quantitative phosphoproteomic analysis of the molecular substrates of sleep need. *Nature* 558, 435–439.

Ximerakis, M., Lipnick, S.L., Innes, B.T., Simmons, S.K., Adiconis, X., Dionne, D., Mayweather, B.A., Nguyen, L., Niziolek, Z., Ozek, C., et al. (2019). Single-cell transcriptomic profiling of the aging mouse brain. *Nat Neurosci* 22, 1696–1708.

Zeisel, A., Muñoz-Manchado, A.B., Codeluppi, S., Lönnerberg, P., Manno, G.L., Juréus, A., Marques, S., Munguba, H., He, L., Betsholtz, C., et al. (2015). Cell types in the mouse cortex and hippocampus revealed by single-cell RNA-seq. *Science* 347, 1138–1142.

REVIEWERS' COMMENTS:

Reviewer #1 (Remarks to the Author):

The authors have satisfied my primary concerns. The manuscript is significantly improved and I think the study significantly contributes knowledge to the field.

Reviewer #2 (Remarks to the Author):

The authors have addressed all outstanding concerns.